# Cortex-wide response mode of VIP-expressing inhibitory neurons by reward and punishment

**Zoltán Szadai**[1,2,3,4†], **Hyun-Jae Pi**[5,6†], **Quentin Chevy**[5,7†], **Katalin Ócsai**[2,4,8,9], **Dinu F Albeanu**[5], **Balázs Chiovini**[1], **Gergely Szalay**[1], **Gergely Katona**[2], **Adam Kepecs**[5,7*], **Balázs Rózsa**[1,4*]

[1]Laboratory of 3D functional network and dendritic imaging, Institute of Experimental Medicine, Budapest, Hungary; [2]MTA-PPKE ITK-NAP B – 2p Measurement Technology Group, The Faculty of Information Technology, Pázmány Péter Catholic University, Budapest, Hungary; [3]János Szentágothai Doctoral School of Neurosciences, Semmelweis University, Budapest, Hungary; [4]BrainVisionCenter, Budapest, Hungary; [5]Cold Spring Harbor Laboratory, Cold Spring Harbor, United States; [6]Volen Center for Complex Systems, Biology Department, Brandeis University, Waltham, United States; [7]Departments of Neuroscience and Psychiatry, Washington University School of Medicine, St. Louis, United States; [8]Computational Systems Neuroscience Lab, Wigner Research Centre for Physics, Budapest, Hungary; [9]Department of Mathematical Geometry, Institute of Mathematics, Budapest University of Technology and Economics, Budapest, Hungary

**\*For correspondence:**
akepecs@wustl.edu (AK);
rozsabal@koki.hu (BR)

[†]These authors contributed equally to this work

**Abstract** Neocortex is classically divided into distinct areas, each specializing in different function, but all could benefit from reinforcement feedback to inform and update local processing. Yet it remains elusive how global signals like reward and punishment are represented in local cortical computations. Previously, we identified a cortical neuron type, vasoactive intestinal polypeptide (VIP)-expressing interneurons, in auditory cortex that is recruited by behavioral reinforcers and mediates disinhibitory control by inhibiting other inhibitory neurons. As the same disinhibitory cortical circuit is present virtually throughout cortex, we wondered whether VIP neurons are likewise recruited by reinforcers throughout cortex. We monitored VIP neural activity in dozens of cortical regions using three-dimensional random access two-photon microscopy and fiber photometry while mice learned an auditory discrimination task. We found that reward and punishment during initial learning produce rapid, cortex-wide activation of most VIP interneurons. This global recruitment mode showed variations in temporal dynamics in individual neurons and across areas. Neither the weak sensory tuning of VIP interneurons in visual cortex nor their arousal state modulation was fully predictive of reinforcer responses. We suggest that the global response mode of cortical VIP interneurons supports a cell-type-specific circuit mechanism by which organism-level information about reinforcers regulates local circuit processing and plasticity.

## Editor's evaluation

The exceptional imaging technique used permitted to detection of the activity of a specific group of cortical neurons known as vasoactive intestinal polypeptide (VIP)-expressing interneurons from several cortical regions with high temporal resolution. The landmark message conveyed by this manuscript is that many VIP-expressing interneurons respond to reward and punishment but also

show regional differences. The conclusions drawn are generally supported by the data. This paper is of potential interest to neuroscientists expert in cortical circuitry and behavioral role of neuron types.

## Introduction

Neocortex contains a number of functionally distinct areas such as the visual, frontal, and motor (Mtr) cortical regions, each specializing in different roles (*Felleman and Van Essen, 1991*). Classical studies have established that the specialization of each region is reflected in their neural responses; for instance, neurons in the visual cortex respond to information about the visual world, while neurons in the Mtr cortex inform about actions. There is an additional layer of mechanisms known to modulate these cortical responses, spanning from the broad effects of arousal to the location-specific impact of attention (*Harris and Thiele, 2011*). Intriguingly, there is also a growing body of evidence suggesting that each area can represent non-classical features such as reward timing (*Monk et al., 2020*) and category representation (*Goltstein et al., 2021*) in visual cortex, visual stimuli and motor modulation in the auditory cortex (ACx; *Attinger et al., 2017*; *Nelson et al., 2013*) and beyond (*Allen et al., 2017*; *Musall et al., 2019*; *Stringer et al., 2019*). Here we pursued a similarly unexpected response pattern based on our previous observation that ACx VIP interneurons respond not only to auditory stimuli but also to reward and punishment (*Pi et al., 2013*).

Vasoactive intestinal polypeptide (VIP) expression demarcates a small interneuron subpopulation (15–20%) located mostly in the upper layers of the cortex (*Kim et al., 2017*; *Staiger et al., 2004*; *Gonchar and Burkhalter, 1997*; *Gonchar et al., 2007*). Previous studies have identified a cortical circuit motif controlled by VIP interneurons that preferentially inhibit other interneurons and thereby disinhibit principal neurons (*Lee et al., 2013*; *Pfeffer et al., 2013*; *Pi et al., 2013*). In this circuit, VIP interneurons mainly inhibit somatostatin interneurons, which tend to exert an inhibitory drive on the dendrites of cortical pyramidal neurons (*Gentet et al., 2012*). Such disinhibition could lead to the selective amplification of local processing and serve the important computational functions of gating and gain modulation (*Pi et al., 2013*). Hence, one proposed role for VIP interneurons is to gate the integration and the plasticity of the synaptic inputs onto pyramidal neurons (*Letzkus et al., 2015*; *Williams and Holtmaat, 2019*). The same stereotyped connectivity was found in functionally and cytoarchitectonically different regions of the brain, across the auditory, prefrontal (*Pi et al., 2013*), visual (*Pfeffer et al., 2013*), and somatosensory (SS; *Gasselin et al., 2021*; *Lee et al., 2013*) cortices, as well as in the amygdala (*Krabbe et al., 2019*) and the hippocampus (*Francavilla et al., 2018*; *Tyan et al., 2014*). However, it is not known whether VIP interneurons have similarly stereotyped functional roles across cortical regions.

VIP interneurons have been shown to have a multiplicity of roles in sensory processing, arousal modulation, learning, and plasticity. First, studies in the primary sensory regions – barrel cortex, ACx, and visual cortex – have demonstrated that tactile, auditory, and visual stimuli drive VIP neuron activity in diverse ways (*Ibrahim et al., 2016*; *Khan et al., 2018*; *Kuchibhotla et al., 2017*; *Mesik et al., 2015*; *Pi et al., 2013*; *Sachidhanandam et al., 2016*). However, the sensory tuning of VIP neurons tends to be weak compared to that of principal neurons. Second, VIP interneuron activity is highly correlated with the changes in pupil dilation and locomotion, suggesting a role in modulating cortical processing across arousal states (*Dipoppa et al., 2018*; *Fu et al., 2014*; *Garcia-Junco-Clemente et al., 2017*; *Jackson et al., 2016*; *Pakan et al., 2016*; *Reimer et al., 2014*; *Zhang et al., 2014*), while other reports show that locomotion modulates sensory processing independently from VIP activation (*Yavorska and Wehr, 2021*). Finally, optogenetic or pharmacogenetic inhibition (*Donato et al., 2013*; *Fu et al., 2015*; *Kamigaki and Dan, 2017*) of VIP interneurons, as well as their developmental dysregulation (*Batista-Brito et al., 2017*; *Fu et al., 2015*) impairs learning and plasticity in sensory discrimination and memory-guided tasks (*Batista-Brito et al., 2017*; *Kamigaki and Dan, 2017*).

We sought to investigate common rules that recruit VIP interneurons. Our starting point was the observation that auditory cortical VIP neurons respond not only to auditory stimuli but also to reward and punishment (*Pi et al., 2013*). VIP cells have been reported to respond to reward in hippocampus and medial prefrontal cortex (mPFC) and to foot shock in amygdala (*Krabbe et al., 2019*; *Pinto and Dan, 2015*; *Turi et al., 2019*). Those later observations fit the function of these areas in learning and plasticity. In contrast, such activity was not observed in the SS and visual cortices (*Khan et al., 2018*; *Sachidhanandam et al., 2016*) calling into question the existence of a global reinforcement-related

VIP interneuron recruitment mode. To address this, we set up to systematically record VIP interneurons across the whole dorsal cortex during an auditory decision task. To allow simultaneous monitoring of large number of VIP interneurons across a variety of cortical regions, we used three-dimensional (3D) acousto-optical (AO) two-photon microscopy, providing both a high signal-to-noise ratio (SNR) and high temporal resolution across large volumes. To gain access to deeper lying cortical regions like mPFC and ACx, we used fiber photometry and measured the population activity of VIP interneurons. We show that most VIP interneurons across cortex are indeed robustly activated by reward and/or punishment, and regional- and task-related behavioral factors contribute to shape their response profile differently. This global mode of recruitment of VIP interneurons is distinct from known arousal modulation of their activity and separate from the local response mode of VIP interneurons.

## Results

### Auditory discrimination task for mice

To probe the behavioral function of VIP interneurons, we trained head-fixed mice (n=22) on a simple auditory discrimination task (*Figure 1A*). Each trial began with the delivery of a 0.5 s auditory stimulus, and mice were trained to lick (go trials) or withhold licking (no-go trials) based on the tone identity. Successful licking after tone delivery during go trials was rewarded with water (Hit trials), while the absence of licking was not rewarded (Miss trials). Licking for no-go trials triggered a mild air-puff punishment (false alarm [FA]), which was omitted if the animal successfully withheld licking (correct rejection [CR]). Mice learned this task over 3 ± 0.6 (mean ± SD) sessions after introducing the no-go tone, reaching a performance level of 80% (percentage of correct responses, Hit or CR). All recordings in this study were obtained early in training to be able to investigate neuronal responses to air puff punishment (FA trials).

### Imaging VIP neurons with fast 3D AO microscopy

To study the reinforcer-mediated dynamics of VIP interneurons across the cortex, we sought to simultaneously record a large number of VIP cells, across a large cortical volume. Because of their sparse cortical distribution, electrophysiological methods combined with optogenetics-assisted identification are less suitable for cortex-wide recordings of VIP interneurons. To overcome this challenge, we used random access, 3D AO two-photon microscopy (*Katona et al., 2012*; *Nadella et al., 2016*; *Szalay et al., 2016*). This method allows to restrict the measurement time solely to the regions of interest (ROIs). Additionally, two-photon fluorescence excitation results in high imaging penetration required for in vivo imaging while also delivering high spatial resolution, therefore limiting neuropil contamination (*Helmchen and Denk, 2005*; *Horton et al., 2013*; *Yildirim et al., 2019*). Here, we used 3D chessboard scanning (*Szalay et al., 2016*) that generates small patches encompassing each neuron soma. This scanning mode preserves fluorescence information during brain movements and thereby allows motion correction in behaving animals (*Figure 1A and C*, for theoretical summary see *Marosi et al., 2019*). Overall, chessboard scanning produces an additional ~170-fold increase in measurement speed and ~15-fold increase in SNR, compared to a high-speed resonant mirror-based system scanning the same volume (Table S1). Thus, we could simultaneously image the activity of up to 120 GCaMP6f-expressing VIP cells (range: 12–120 cells) in a 689 μm×639 μm×580 μm scanning volume at a minimum of 27.8 Hz rate (*Figure 1B and C*).

### VIP neurons are simultaneously activated by reward and punishment in parietal cortex

We first focused on measuring the calcium-related activity of VIP interneurons in the medial parietal association area (MPta). *Figure 1* shows an example recording of 120 VIP interneurons from the MPta while the animal performed the auditory discrimination task described above. We found that the majority of VIP interneurons responded to reward and punishment presentation (reward = 85%, punishment = 90%, reward and punishment = 75% of recorded VIP interneurons). Individual neurons showed a high reliability in their recruitment (percentage of active trials for a given neuron) of 64 and 77% for Hit and FA trials, respectively. Examining individual trials, 49 and 60% of VIP interneurons were simultaneously activated by reward and punishment, respectively (*Figure 1D*, synchronicity). On the contrary, parvalbumin (PV) interneurons, another inhibitory neuron cell type, did not show

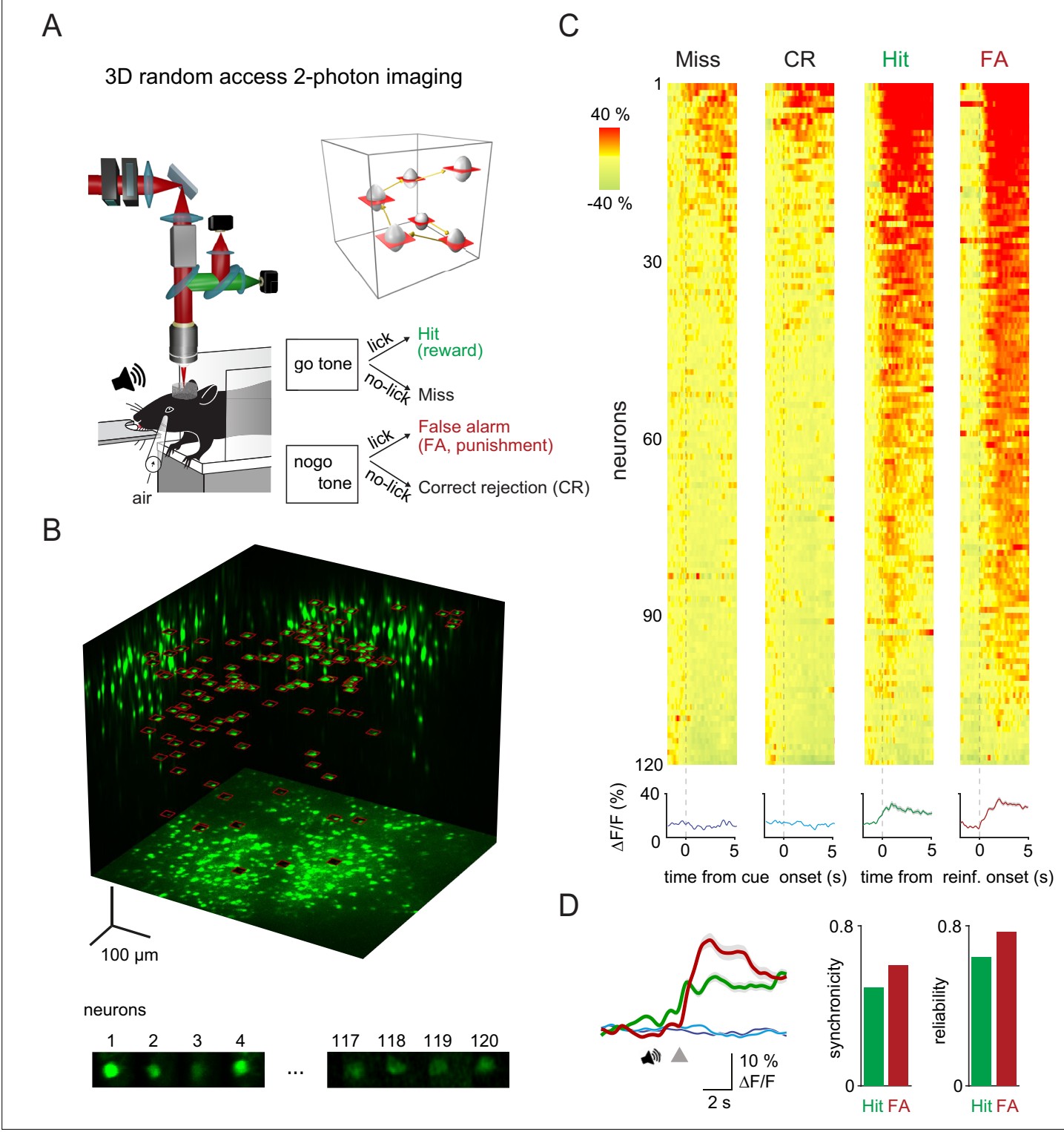

**Figure 1.** Three-dimensional (3D) random access two-photon imaging of vasoactive intestinal polypeptide (VIP) neurons in an auditory discrimination task. (**A**) Schematic of the combined fast 3D acousto-optical (AO) imaging and behavior experiments. Head-restrained mice were trained to perform a sensory discrimination, an auditory go-no-go task during 3D AO imaging using the chessboard scanning method (inset). (**B**) Maximal intensity 3D projection of the GCaMP6f-labeled VIP interneuron population imaged by fast 3D AO scanning in the medial parietal association area. All 120 neurons within the cubature were simultaneously imaged using tiled chessboard scanning (red squares). Bottom shows tile examples containing single-neuron somata obtained using chessboard scanning. Note that some background was included to allow for motion correction. (**C**) Top, example somatic Ca²⁺ responses recorded of neurons recorded as in (**B**) during Miss, correct rejection (CR), Hit, and false alarm (FA) trials. Responses were ordered according

*Figure 1 continued on next page*

*Figure 1 continued*

to their maximum amplitude for each trial types. Traces were aligned to cue onset for Miss and CR trials and to reward or punishment delivery for Hit and FA. Bottom, neuron average response for each trial type (mean ± SEM). (**D**) Left, average transients of a measurement session (128 trials) for Hit (green), FA (red), Miss (dark blue), and CR (light blue) responses recorded from the 120 VIP interneurons. Gray triangle marks the reinforcement onset in case of Hit and FA. Averages of Miss and CR trials were aligned according to the expected reinforcement delivery calculated based on the average reaction time. Right, average synchronicity (mean ± SEM) and trial-to-trial repeatability (reliability) of individual neuronal responses.

The online version of this article includes the following video and figure supplement(s) for figure 1:

**Figure supplement 1.** Three-dimensional random access two-photon imaging and fiber photometry of vasoactive intestinal polypeptide (VIP) neurons in an auditory discrimination task.

**Figure 1—video 1.** Recording sparse interneuronal population in large volume.

https://elifesciences.org/articles/78815/figures#fig1video1

**Figure 1—video 2.** Vasoactive intestinal polypeptide (VIP) population activity during an auditory discrimination task.

https://elifesciences.org/articles/78815/figures#fig1video2

comparable homogeneity in their recruitment by primary reinforcers. Reward and punishment delivery induced an increase in activity of, respectively, 29 and 10% of PV interneurons recorded in MPta (*Figure 1—figure supplement 1F*).

## VIP neurons are activated by reward and punishment across dorsal cortex

We then extended recordings of VIP interneurons to most of dorsal cortex including visual, SS, Mtr, and parietal areas (*Figure 2A, B*, 16 mice, one or two areas per mouse). Among the 811 neurons imaged in 18 imaging sessions from 16 mice, 65 VIP interneurons did not show statistically significant responses to any behavioral events (e.g. auditory or visual stimulation, reward or punishment delivery) and were therefore excluded from further analyses. 83 and 85% of the remaining 746 VIP interneurons, responded to reward and punishment, respectively (*Figure 2C, D*). We found that 73% of the VIP interneurons significantly responded to both reward and punishment, similar to our observations in MPta. Only 15% of the VIP interneurons responded to auditory cues in Miss and CR trials (*Figure 2C, D*). 12% of the cells were activated in all trial types. The response of VIP interneurons to reward and punishment showed a strong correlation (Pearson correlation coefficient for average amplitudes: 0.73, *Figure 2E*). High synchronicity and reliability in VIP interneuron recruitment were also observed in our recordings extending throughout the dorsal cortex: on a given trial, 58% of VIP interneurons were simultaneously activated (57 ± 2.4% and 58 ± 2.5%, for Hit and FA trials, respectively, *Figure 2—figure supplement 1D*) with a reliability of 61% (59 ± 1.7% and 63 ± 2.4%, for Hit and FA trials, respectively, *Figure 2—figure supplement 1C*). The stability of the responses was maintained across trials (*Figure 2—figure supplement 1E*).

## Recruitment of VIP interneuron population by reward and punishment in the medial prefrontal and auditory cortex

To probe additional deep lying cortical structures, we took advantage of the coherent recruitment of VIP interneurons by reinforcers and used fiber photometry (*Cui et al., 2013*; *Gunaydin et al., 2014*). This approach allowed us to simultaneously measure bulk calcium-dependent signals from VIP interneurons located in the right mPFC and left ACx by implanting two 400-µm optical fibers at these locations (n=6 sessions from n=6 mice, *Figure 1—figure supplement 1C*). Consistent with our previous electrophysiological results in ACx (*Pi et al., 2013*) and two-photon imaging from dorsal cortical regions, calcium-related signals from VIP interneurons in the ACx and mPFC were increased after reward and punishment delivery (in ACx: Hit = 4.8 ± 0.32%, FA = 10.9 ± 0.03%; in mPFC: Hit = 4.3 ± 0.69%, FA = 6.6 ± 0.85%, ΔF/F peaks, *Figure 2A*). We did not further analyze the FA responses in ACx as those responses also had a sensory component reflecting the white noise-like sound created by the air puff delivery. Because the cue delivery could prove as a confound to measure reward-mediated responses from VIP interneurons in ACx (see also methods), we also delivered random rewards in separate sessions. Water drop delivery recruited VIP interneurons in both ACx and mPFC in a similar fashion to water delivery during the discrimination task (*Figure 2—figure supplement 1G*). Like our single-cell results, the PV-expressing interneuron population in ACx did not show any

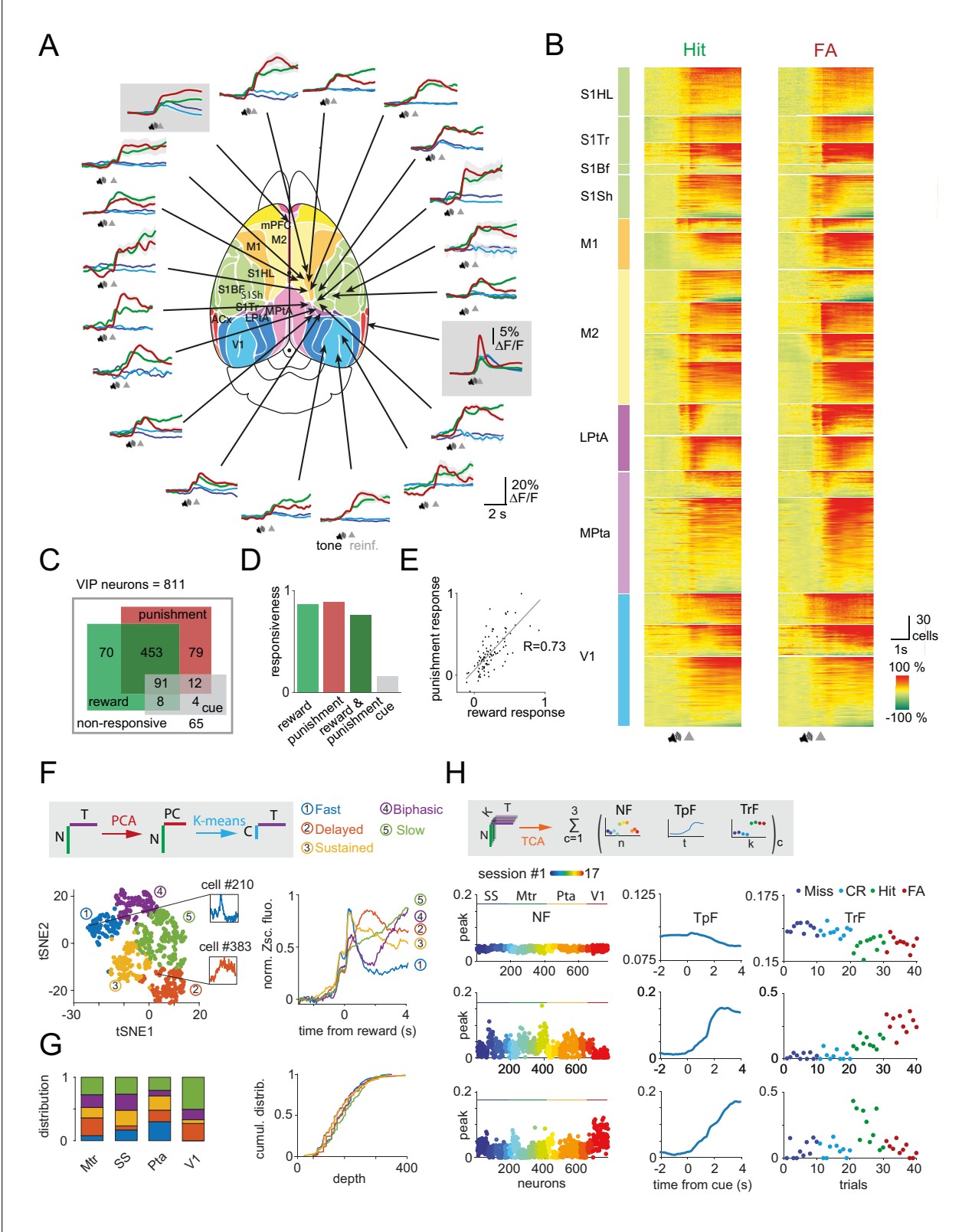

**Figure 2.** Reward and punishment recruit vasoactive intestinal polypeptide (VIP) neuronal activity across the dorsal cortex (**A**) Ca²⁺ responses of individual VIP interneurons recorded separately from 18 different cortical regions from 16 mice using fast three-dimensional acousto-optical imaging were averaged for Hit (thick green), false alarm (FA; thick red), Miss (dark blue), and correct rejection (CR; light blue). Fiber photometry data were recorded simultaneously from medial prefrontal cortex (mPFC) and auditory cortex (ACx) regions and are shown in gray boxes. Functional map

*Figure 2 continued on next page*

*Figure 2 continued*

(*Pankhurst et al., 2012*) used with the permission of the author. Speaker symbols represent the average time of tone onset, and gray triangles mark the reinforcement onset for Hit and FA. Averages of Miss and CR trials were aligned according to the expected reinforcement delivery calculated on the basis of the average reaction time. The auditory discrimination task in case of fiber photometry measurements was designed such that if a mouse licks during the tone, the tone stops immediately, and reinforcers are delivered. Note the different scalebar for photometry measurements. mPFC (n=6 mice), ACx (n=6), S1Hl/S1Tr/S1Bf/S1Sh: primary somatosensory cortex, hindlimb/trunk/barrel field/shoulder region (n=4), M1/M2: primary/ secondary motor cortex (n=6), Mpta/Lpta: medial/lateral parietal cortex (n=4), V1: primary visual cortex (n=3). (**B**) Each line of the raster plots shows average neuronal response for Hit and FA. Responses were aligned to reinforcement onset before averaging. Abbreviations indicate color-coded cortical recording positions shown in panel A. Speaker symbol represents the approximate time of tone onset, as reaction times of the animals could be different. Responses were normalized in each region and ordered according to their maximum amplitude. (**C**) Responsiveness of 811 VIP interneurons for Hit and FA. (**D**) Bar chart of data from C. (**E**) Average response of individual VIP interneurons for FA as a function of the response for Hit. Note the high correlation ($R$=0.73). (**F**) Left, T-distributed stochastic neighbor embedding (tSNE) plot of the reward-mediated activity of VIP interneurons after principal component analysis (PCA). Individual neurons are color coded according to their cluster type obtained using a k-means clustering algorithm. Inserts show the average response of single rapidly (blue) or delayed (orange) activated VIP interneurons. Right, average GCaMP6f responses from different clusters of VIP interneurons after reward delivery. N: number of neurons, T: time, PC: principal components, C: clusters. (**G**) Left, distribution of the clusters shown in panel F across different cortical areas (Mtr: motor cortex, SS: somatosensory cortex, Pta: parietal cortex, V1: primary visual cortex). Right, cumulative distribution of the clusters shown as a function of cortical depth. (**H**) Top, schematics of temporal component analysis: single trial neuronal data were decomposed in a sum of latent components. N: number of neurons coming from different sessions, K: trial numbers, and T: time. Below, rank 3 tensor component analysis (TCA) neuron (NF), temporal (TpF), and trial (TrF) factors. Miss and CR trial factors were indicated here with dark and light blue dots. The second component clearly distinguishes between trials with and without reinforcement.

The online version of this article includes the following figure supplement(s) for figure 2:

**Figure supplement 1.** Quantification of the activity of vasoactive intestinal polypeptide (VIP) neurons across the dorsal cortex.

**Figure supplement 2.** Heterogeneity in vasoactive intestinal polypeptide (VIP) neuronal responses across the dorsal cortex.

**Figure supplement 3.** Cell diameter distribution of the recorded vasoactive intestinal polypeptide (VIP) interneuron population and its relation to the activity.

---

significant change in activity upon similar random reward delivery (*Figure 2—figure supplement 1G*). Concurrent recordings of VIP interneuron populations in ACx and mPFC revealed heterogeneity in the dynamics of VIP interneuron activity during reward delivery (*Figure 2A*). VIP interneurons in the ACx showed phasic response to reward (peak time for Hit = 0.06 ± 0.036 s, decay time constant = 2.7 s). In contrast, medial prefrontal VIP interneurons were slowly activated (peak time for Hit = 3.08 ± 0.968 s, decay time constant = 7.75 s, *Figure 2A*). These population recordings confirmed the dominant contribution of reinforcement-related signals to VIP interneuron population responses and also revealed potential area-specific heterogeneity in the dynamics of VIP interneuron activity.

## Heterogeneity in the dynamics of reinforcer-related activity of individual VIP interneurons

The difference in dynamics at the population level across different brain areas might be supported by heterogeneity in the individual response profiles of VIP interneurons. Thus, we sought to characterize the dynamics of VIP interneurons at a single-cell resolution and across dorsal cortex using chessboard 3D AO recordings. We first focused on VIP interneurons activated upon reward delivery during the sensory discrimination task (n=606 cells). We applied principal component analysis (PCA) to the average reward responses of individual neurons to reward. We then clustered these responses using k-means clustering. This approach did not readily separate neurons according to the recording sessions (*Figure 2—figure supplement 2A*). Rather, our clustering approach allowed us to delineate five functional groups of VIP interneurons (*Figure 2F*). Based on visual inspection of their mean temporal profiles, we labeled these groups as: 'fast' (n=109), 'delayed' (n=88), 'sustained' (n=177), 'biphasic' (n=120), and 'slow' (n=112). Note that all of these response types share important similarities such as a phasic reward response and mostly differ in their subsequent temporal dynamics. We first considered the distribution of these five types of neuronal responses across different brain areas. We observed an overrepresentation of the 'fast' group in parietal cortex and of the 'slow' group in primary visual cortex (*Figure 2G*). The 'fast' group was absent from visual cortex (*Figure 2G*). To quantify this heterogeneity across cortical areas, we defined five feature vectors as the mean response profile of each cluster to rewards and then projected the reward response of each VIP interneuron onto these features (*Figure 2—figure supplement 2B*). We found that the

projection associated with the 'fast' group was significantly higher for VIP interneurons located in parietal compared to those recorded in visual cortex (mean $\Delta_{Pta-V1}$=3.22, Mann-Whitney test, p=3.77 $10^{-9}$), while the opposite was observed for the projection associated with the 'slow' group (mean $\Delta_{Pta-V1}$=−7.79, p=3.77 $10^{-9}$, *Figure 2—figure supplement 2B*). Finally, we took advantage of the 3D AO imaging to investigate the heterogeneity in the responses of VIP interneurons located at different depth of the cortex. We were able to detect some differences in the amplitude of the average responses for reward (*Figure 2—figure supplement 1F*, *F*=9.5, p=0.002). However, we did not observe any differences in the distribution of the different clusters across depth (*Figure 2G*, *F*=1.16, p=0.36).

The groups delineated by the PCA-based cluster analysis possibly represent distinct subtypes with variable genetic and connectivity patterns. We addressed subtype diversity by measuring the diameter (full width at half maximum [FWHM]) of the somata and found bimodal distributions both at the level of individual mice (7/8 mice) and also the entire population examined ($R^2$=97.7%, *F*=206.4, p<0.001, n=8 mice n=439 cells, *Figure 2—figure supplement 3A*). This separation might reflect distinct VIP subtypes categorized based on the expression of calretinin and cholecystokinin (*Luo et al., 2020*; *Staiger et al., 2004*; *Gonchar and Burkhalter, 1997*). In contrast, the soma diameter of the responsive and non-responsive neurons was similar (Hit: 10.0 ± 0.1 vs 9.9 ± 0.1 µm, p=0.39, FA: 10.3 ± 0.2 vs 9.8 ± 0.1 µm, p=0.07, Mann-Whitney test, *Figure 2—figure supplement 3B*). We next considered whether there are differences across functional groups identified using PCA-based clustering. VIP-fast and VIP-sustained groups had somewhat larger cell bodies (fast: 10.2 ± 0.4 µm, delayed: 9.8 ± 0.2 µm, sustained: 10.4 ± 0.2 µm, biphasic: 10.0 ± 0.2 µm, slow: 9.4 ± 0.2 µm, *Figure 2—figure supplement 3C*), further supporting the possibility that some of the identified functional VIP groups correspond to distinct genetic VIP interneuron subtypes.

The differences in average response dynamics from individual neurons could also arise from differential variability across trials. To evaluate this potential heterogeneity in the single-trial dynamics of VIP interneuron activity, we used tensor component analysis (TCA, *Figure 2H*). TCA allowed us to further characterize the trial identity-dependent dynamics of VIP interneuron activity. All recorded neurons from different sessions/cortical regions were grouped by keeping only the first 10 trials of each trial types (see Methods for additional information). After preprocessing (*Figure 2—figure supplement 2C*), we used non-negative tensor decomposition (*Kolda and Bader, 2009*; *Williams et al., 2018*) and focused our analysis on rank 3 TCA as using higher rank showed signs of overfitting and did not improve the reconstruction error (22% for rank 3 vs 18% for rank 20). We found a latent temporal component that robustly separated Hit and FA trials from Miss and CR trials (second component, *Figure 2H*, mean$_{Miss\&CR}$ vs mean$_{Hit\&FA}$=0.02 vs 0.2, Mann-Whitney test, p<0.001). The third latent temporal component showed a slower time course, with some reward specificity and was overrepresented in neurons from the visual cortex (mean$_{SS,Mtr,Pta}$ vs mean$_{V1}$=0.02 vs 0.06 p<0.001). TCA allowed us to quantify on the one hand, the brain-wide recruitment of VIP interneurons by reward- and punishment-associated responses and, on the other hand, brain region-dependent heterogeneity in dynamics (e.g. slower dynamics in the visual cortex) in an unbiased manner.

Beyond neuron-to-neuron and inter-trial variability, inter-individual (animal-to-animal) variability can also contribute to the heterogeneity experienced in the responses. Therefore, we calculated the coefficient of variation (CV = SD/mean) of the peak of responses among the neurons of individual recordings (cell-to-cell) and compared the result to the CV calculated using the average peak amplitudes of the sessions (inter-individual). We conducted this analysis in parietal cortical measurements (n=4 mice, n=236 cells) separately in the four trial types (*Figure 2—figure supplement 2E*). The cell-to-cell CV was comparable among the trial types, Hit and FA being slightly less variable (Miss: 0.80, CR: 0.70, Hit: 0.57, FA: 0.58). The inter-individual variability in Miss and CR was similar in magnitude to the neuron-to-neuron variability, but in Hit and FA, it was surprisingly, ~2.6-fold lower (Miss: 0.94, CR: 0.69, Hit: 0.25, FA: 0.20). This reveals that while responses to the sound cue (without reinforcement) were as variable at the level of individual mice as at the level of individual cells, the reinforcement-related response was less variable across animals, and differences rather came from cellular differences. This agrees with the PCA-based clustering results that showed no separation according to mice and imaging sessions (*Figure 2—figure supplement 2A*).

## Behavioral performance influences task-related VIP interneuron responses

Differences in individual animal performance of the discrimination task could also contribute to the heterogeneity in the activity of VIP interneurons. Hence, we tested whether differences in hit rate influenced the response of VIP interneurons during various epochs. We observed a positive correlation between the hit rate and the magnitude of the cue response of VIP interneurons during Hit trials ($R=0.62$, *Figure 2—figure supplement 2D*). Using a simple linear regression model, we found that the hit rate was able to explain 39% of the variance of cue responses ($R^2=39.0\%$; $p=0.006$). For comparison, the cue response was not influenced by the location of the imaging site (visual, SS, Mtr or parietal) from where the activity of the VIP interneurons was recorded ($R^2=18.2\%$; $p=0.41$).

## Arousal modulation of reinforcement-mediated recruitment of VIP interneurons

Reward and punishment can induce changes in the arousal states of the animals, and the activity of VIP interneurons is known to be modulated by the arousal states (*Fu et al., 2014*; *Garcia-Junco-Clemente et al., 2017*; *Reimer et al., 2014*). Therefore, we considered whether changes in arousal contributed to the recruitment of VIP interneurons by primary reinforcers. We first monitored variations in pupil diameter as a proxy for assessing arousal states (*Vinck et al., 2015*). In this set of experiments, we restricted our measurements to SS and Mtr cortices using 3D AO microscopy and to the ACx and the mPFC using fiber photometry as described above (*Figure 3A*).

Hit and FA trials were first split into two groups using the mean reinforcer-mediated pupil dilation for threshold (average changes in pupil size for the large and small pupil group: 14.38 vs 2.81%; *Figure 3C*). Reinforcement delivery associated with larger changes in pupil diameter led to a stronger recruitment of VIP interneurons in both the SS (large vs small pupil ΔF/F: 40 vs 29% n=26, t-test, p=0.01) and Mtr cortex (large vs small pupil ΔF/F: 28 vs 18%, n=111, p<0.001; *Figure 3D*). This modulation was strongest in the late response phase (difference between large vs small pupil associated ΔF/F at initial [0–1 s] and late [2–3 s] phase: 10.7 ± 0.01 vs 19.9 ± 0.01%, p<0.001 *Figure 3—figure supplement 1E*). A comparable modulation of VIP interneuron activity upon reinforcer presentation was observed when trials were split based on baseline pupil diameter (see Supplemental information and *Figure 3—figure supplement 1B*). In a subset of trials with high baseline arousal we found that reinforcement-related signals in VIP neurons were not associated with further increase in arousal (Hit ΔF/F: 4.4 ± 1.1% in 14.7% of the trials, t-test, p<0.05, n=4 mice, FA ΔF/F: 20.4 ± 0.8%, in 11.4% of the trials, p<0.01, n=3 mice, *Figure 3—figure supplement 2A–C*). The recruitment of VIP interneurons upon cue presentation only (i.e. Miss and CR trials, where trials were split using the mean cue-mediated pupil dilation for threshold) was similarly modulated by arousal (*Figure 3—figure supplement 1A*). The positive correlation between pupil size changes and reinforcer-related activity of VIP interneurons was also present at single-cell level (median correlation coefficient 0.31). Interestingly, neurons with a larger correlation coefficient showed a slower activity profile of recruitment by reinforcers than those with a correlation coefficient below the median value (*Figure 3B*).

Because VIP interneuron population had slower dynamics in the mPFC than in the ACx, we hypothesize that the pupil size-dependent modulation of reward responses would be stronger in prefrontal cortex than in ACx. Indeed, reinforcer-mediated responses in mPFC were significantly larger in trials with greater changes in pupil diameter (large vs small pupil ΔF/F: 3.8 vs 1.4%, n=6 mice, p=0.01; *Figure 3E* and *Figure 3—figure supplement 1C*). This pupil size-dependent modulation was, however, absent in recordings of VIP interneurons in the ACx (large vs small pupil ΔF/F: 4.5 vs 3.2%, n=6 mice, p=0.08 in the ACx; *Figure 3—figure supplement 1A*). Similarly to our single neuron measurements, arousal modulation was present at the population level in Miss and CR trials (*Figure 3—figure supplement 1A*) or when using the baseline pupil dilation as arousal index, except for the ACx (*Figure 3—figure supplement 1B*). Overall, while we found arousal-related boosting of most response types, the fast recruitment of VIP interneurons by reinforcers was observed independently of the arousal state of the animal (high/low) at baseline or during reward/punishment delivery.

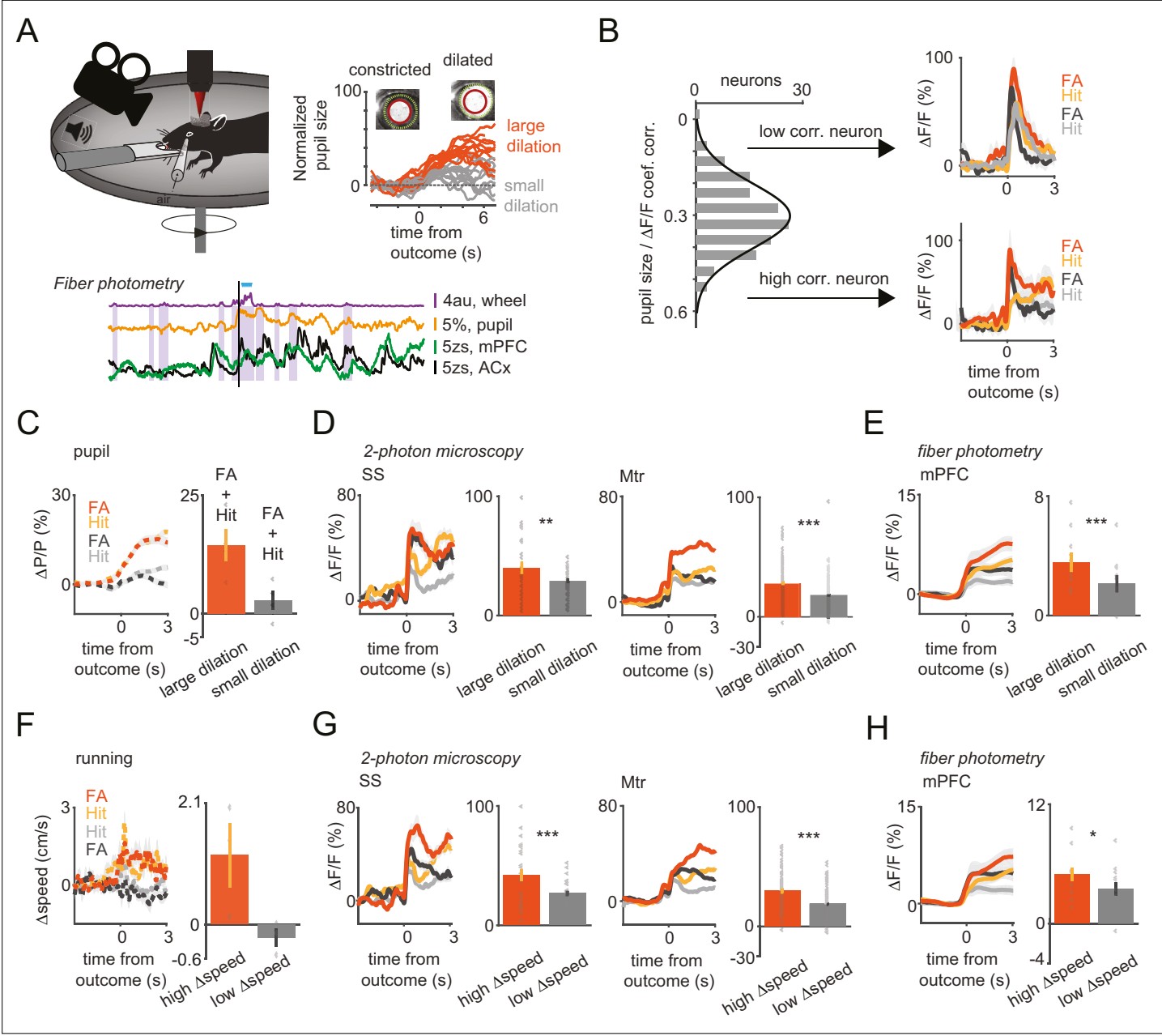

**Figure 3.** Arousal states modulate vasoactive intestinal polypeptide (VIP) neural responses to sensory cues and reinforcers. (**A**) Upper left, schematic of measurements. Pupil and movement were simultaneously monitored during three-dimensional (3D) imaging in the auditory go-no-go task. Upper right, high (orange) and low (gray) arousal states were separated by changes in pupil diameter. Below, 60 s continuous monitoring of different behavioral variables together with VIP interneuron population activity in auditory cortex (ACx) and medial prefrontal cortex (mPFC). The black bar indicates the timing of an uncued reward delivery. Blue triangles indicate licking events. Purple-shaded boxes represent running bouts. (**B**) Left, distribution of correlation coefficients of relative change in pupil diameter (ΔP/P) and VIP neuronal response. Right, reinforcement-associated responses were significantly larger when relative change in pupil diameter (ΔP/P) was higher during the task. Red and orange indicate false alarm (FA) and Hit responses associated with higher ΔP/P. FA and Hit responses associated with low ΔP/P are in black and gray, respectively. (**C**) Average pupil dilation traces during high (red and orange) and low (black and gray) pupil changes for FA and Hit trials for somatosensory (SS) and motor (Mtr) recordings in panel D. Bars indicate average amplitudes (mean ± SEM, Hit and FA combined). (**D**) Population averages for Hit and FA during high and low pupil change in the SS and Mtr regions. Bars indicate average amplitudes (mean ± SEM, Hit and FA combined). Even in the late period, when the outcome responses were dissipated, larger changes in pupil diameter at the time of reinforcement were associated with higher VIP responses. (**E**) Same as D but for fiber photometry in the mPFC. Corresponding pupil dilation traces can be found in *Figure 3—figure supplement 1*. (**F**) Same as C but for running speed. (**G**) Same as D but for running speed. (**H**) Same as E but for running speed. Higher relative change in the running speed was associated with larger neuronal responses recorded with 3D imaging or fiber photometry.

*Figure 3 continued on next page*

*Figure 3 continued*

The online version of this article includes the following figure supplement(s) for figure 3:

**Figure supplement 1.** The baseline and the change in pupil diameter, and the change of speed additionally modulate vasoactive intestinal polypeptide neuronal activity on top of activation by cues and outcomes.

**Figure supplement 2.** Arousal and reinforcement can make distinct contributions to vasoactive intestinal polypeptide (VIP) interneuronal activity.

## Modulation of reinforcement-mediated recruitment of VIP interneurons by locomotion

We next examined whether running behavior modulates VIP neuron activity. Response profiles were split based on the median speed change during reinforcer delivery (average speed change for the two groups: 1.20 cm/s vs –0.23 cm/s *Figure 3F*). Similar to what we observed with the pupil size, VIP interneuron responses to reward were stronger when the mice ran faster both for SS (high vs low speed change ΔF/F: 42 vs 27% n=26, t-test, p<0.001) and Mtr cortex-located interneurons (high vs low speed change ΔF/F 30 vs 19%, n=86, p<0.001, *Figure 3G*). A similar difference was also found for sensory responses during Miss and CR (*Figure 3—figure supplement 1D*). However, the correlation was weaker at the single-cell level (median correlation coefficient, 0.11) compared to the pupil modulation. VIP interneuron population activity was modulated by running speed in mPFC (high vs low speed change, ΔF/F: 5 vs 3.5%, p=0.02) but not in ACx (ΔF/F: 5.4 vs 4.1%, p=0.44) similar to what found for pupil modulation (*Figure 3H* and *Figure 3—figure supplement 1D*).

Finally, to quantify the motor- and reinforcement-related contributions to VIP interneuron activity, we built a generalized linear model using the behavior and imaging data from the SS and Mtr recordings (*Figure 3—figure supplement 2D*, n=3 mice). This model was able to explain 18.8 ± 11.1% of the variance of the VIP population calcium signal and highlighted that arousal was the best predictor, followed by reward, punishment, locomotion velocity, and auditory cue (weights = 0.055, 0.031, 0.028, 0.020, and 0.018, respectively; all predictors, except the auditory cue in the case of one animal contributed significantly, p<0.001). These analyses further underscore that running and arousal changes alone cannot fully explain the recruitment of VIP interneurons by reinforcers.

## Comparison between local and global recruitments of VIP neurons in visual cortex

VIP interneurons in ACx and visual cortex respond to sensory input, albeit their responses are weakly tuned (*Kerlin et al., 2010*; *Mesik et al., 2015*; *Pi et al., 2013*). Therefore, we evaluated how local, sensory-evoked responses compared to global, reinforcement-evoked activity of the VIP interneurons in visual cortex. In this set of experiments, to ensure stimulus control (head and body movement) and limit arousal changes, we lightly anesthetized mice with isoflurane and imaged responses to drifting grating bars with different orientations. The vast majority of VIP interneurons (93.5%) responded to visual stimuli. We computed the orientation selectivity index (OSI) and direction selectivity index (DSI) of each VIP neuron (see Methods). As previously reported (*Kerlin et al., 2010*; *Mesik et al., 2015*), VIP interneurons showed broad tuning with little or no preferred directions or orientation (OSI = 0.17 ± 0.01, DSI = 0.16 ± 0.01, *Figure 4C*) especially compared to pyramidal cells (OSI = 0.63 ± 0.01, t-test, p<0.001, DSI = 0.34 ± 0.01, p<0.001).

After measuring their visual response tuning and removing of the light anesthesia, we imaged the same visual cortical neurons while mice performed the auditory discrimination task (*Figure 4A*). We found that 80.4% of VIP neurons were significantly activated by reward or punishment with a response magnitude comparable to their visual responses. The reinforcement-evoked responses were only weakly correlated with visual stimulus-evoked responses (Pearson's R value for reward: 0.16, for punishment: 0.23, and reward and punishment combined: 0.22, $R^2$=0.05, *F*=8.3, p=0.005; *Figure 4D* and *Figure 4—figure supplement 1A*). Neither the orientation nor the direction selectivity of the VIP neurons correlated with their reinforcement responses (Pearson's R value for OSI: 0.08, p=0.31, for DSI: 0.06, p=0.45; *Figure 4—figure supplement 1B*). This supports the hypothesis that the global recruitment of VIP interneurons by reinforcers arises independently of the local circuit processing these neurons might be involved in.

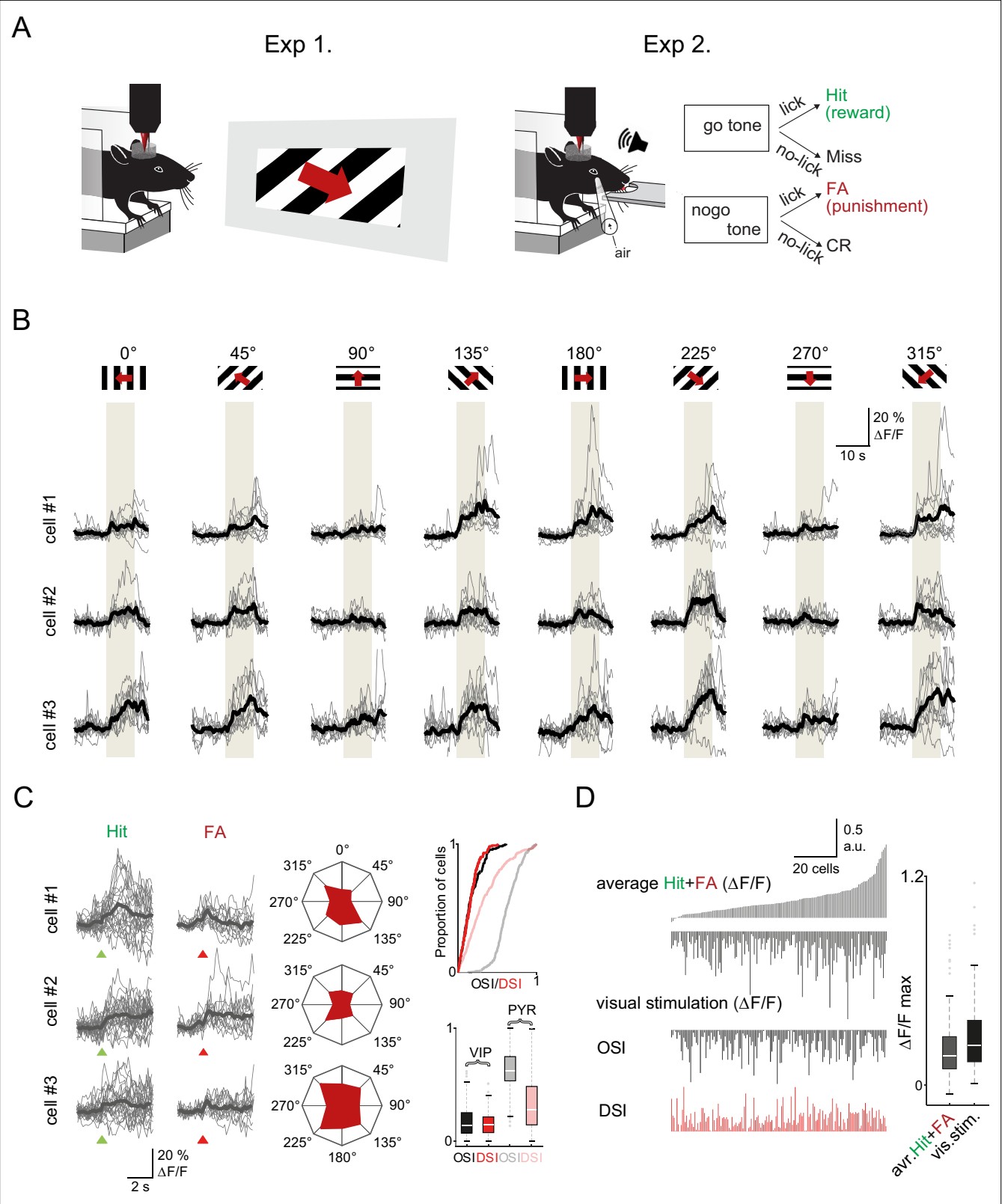

**Figure 4.** Visual cortex vasoactive intestinal polypeptide (VIP) neurons respond to both visual stimuli and reinforcers (**A**) Schematic of the measurement. Orientation tuning was mapped in a first set of experiments (Exp. 1) which was followed by recordings of the same neurons during the auditory go-no-go task (Exp. 2). Both set of recordings were performed using fast three-dimensional acousto-optical imaging. (**B**) Individual Ca²⁺ responses from three different VIP interneurons to visual stimulation with moving grating in eight different directions. The gray boxes indicate the duration of the visual

*Figure 4 continued on next page*

*Figure 4 continued*

stimulation. (**C**) Left, responses of the same three cells to reinforcement. Middle, polar plots of neuronal responses to visual stimulation from the same neurons. Right top, cumulative distribution plot of orientation selectivity index (OSI) and direction selectivity index (DSI) parameters of VIP (black and red) and pyramidal cells (gray and pastel red), (VIP: n=157 cells, n=3 mice, pyramidal cells: n=383 cells, n=3 mice). Right bottom, OSI and DSI values of the same cells. Box-and-whisker plots show the median, 25th and 75th percentiles, and range of nonoutliers and outliers. (**D**) Correlation between reinforcement and visual responses in the same VIP interneurons (n=157). Each column refers to a single cell. From top to bottom: mean of the average Hit and false alarm (FA) responses, average visual responses, mean OSI, and mean DSI. The cells were ordered according to the amplitude of the averaged reinforcement signal. Right, maximums of reinforcement-related and visual stimulation responses. Box-and-whisker plots show the median, 25th and 75th percentiles, and range of nonoutliers and outliers.

The online version of this article includes the following figure supplement(s) for figure 4:

**Figure supplement 1.** Quantification of the connection of visual tuning parameters and reinforcement-related responses.

## Discussion

Here we examined the rules of behavioral recruitment for VIP interneurons across neocortex. By monitoring VIP neural activity across dozens of cortical regions, we found that most neurons were strongly activated by water reward and air-puff punishment. This recruitment was boosted during high arousal states as previously observed for other sensory-mediated processes (*Fu et al., 2014*; *Garcia-Junco-Clemente et al., 2017*; *Reimer et al., 2014*). In visual cortex, the sensory tuning to visual gratings was not predictive of the VIP interneuron responses to reinforcers demonstrating the co-existence of both local and global activation modes.

VIP interneurons represent a small and sparsely distributed population across cortex rendering their investigation challenging (*Kim et al., 2017*; *Gonchar and Burkhalter, 1997*). We monitored the activity of a large population of VIP interneurons in behaving mice across the dorsal cortex. This was made possible by the use of 3D AO two-photon microscopy. The chessboard scanning method of 3D AO microscopy provided additional advantages to our ability to measure the spatial and temporal dynamics of VIP interneuron activity. According to our calculation, this method leads to several orders of magnitude increase in the measurement speed and SNR compared to piezo-based volume scanning (*Supplementary file 1*, Table S1). Furthermore, it enabled robust off-line motion correction during behavioral experiments owing to the ability to actively extend the recordings beyond the soma of the neurons, thereby preserving fluorescence information during motion (*Reid et al., 2016*). Due to this large improvement in the SNR and recording speed (*Supplementary file 1*, Table S1), we were able to dramatically increase the number of simultaneously recorded neurons while maintaining a high sensitivity of detection of neuronal activity. This allowed us to demonstrate that VIP interneurons, throughout cortex and across cortical layers, responded homogenously and synchronously to reward and punishment delivery.

Several previous reports have shown that VIP interneurons can be activated by behavioral reinforcers in the ACx (*Pi et al., 2013*), the mPFC (*Pinto and Dan, 2015*), the hippocampus (*Turi et al., 2019*) and the amygdala (*Krabbe et al., 2019*). Since interneurons are generally strongly connected with cortical circuitry and participate in region-specific local computations, it could be argued that in many of these cases VIP interneuron response properties fit the learning and plasticity roles of the relevant areas. Other reports about the behavioral responses of VIP interneurons in SS and visual cortices failed to observe reinforcement responses. This discrepancy may be explained by experimental differences, as these studies were done in highly trained animals, no air puff punishment was applied and reward delivery was fully predictable. In line with these possibilities, in our measurements from highly trained animals, the cue component dominated the signal (*Figure 2—figure supplement 2D*).

Our observations also revealed heterogeneity in the temporal response of VIP interneurons to reinforcers. In addition to the single-trial and individual neuronal variability in the dynamics of reinforcer-related activity revealed by principal and tensor component analyses, we identified variability in behavioral performance as a source of heterogeneity in the cue-mediated recruitment of VIP interneurons (*Figure 2—figure supplement 3D*). We found some heterogeneity across cortical regions in VIP interneuron responses. For instance, VIP interneurons showed a faster recruitment by reward in ACx than in mPFC (*Figures 2A and 3E* and *Figure 3—figure supplement 1A*). Rapidly activated neurons were absent from visual cortical area whereas they could be observed throughout the rest of the dorsal cortex (*Figure 2G*). This variability might partially reflect different VIP interneuron subtypes

(*Prönneke et al., 2015*; *Staiger et al., 2004*; *Gonchar and Burkhalter, 1997*). Perhaps the most distinct subclass of VIP interneurons is cholecystokinin (CCK+)-expressing interneurons with basket cell-type morphology (*He et al., 2016*). These cells provide perisomatic inhibition in the hippocampus (*Tyan et al., 2014*). We expect that the majority of the neurons we recorded in the upper layers are calretinin (CR+)-expressing bipolar cells, including intrinsic cholinergic acetyltransferase (ChAT+)-positive neurons (*Kim et al., 2017*). This separation into CCK-, CR-, or ChAT-expressing VIP interneurons has been recently partially validated using single-cell transcriptomic analysis (*Tasic et al., 2016*; *Zeisel et al., 2018*; *Paul et al., 2017*).

We identified five functional clusters of VIP interneurons based on differences in their temporal response dynamics. These VIP groups also showed small differences in their cell body diameter, which points to the possibility that genetically different VIP subtypes underlie response heterogeneity. Nevertheless, given the high proportion of VIP neurons responding to reward and punishment, multiple or even all VIP subtypes are expected to have this global response mode. Subtype-dependent differences in VIP activity were mostly present in the late response phase. Further studies using intersectional targeting strategies will be required to provide insight for the potential cell-type-specific origins of the heterogeneity in reinforcement responses.

The response heterogeneity of the local mode of VIP interneurons had already been appreciated for sensory stimuli. When local activation is examined in terms of tuning, VIP interneurons are significantly more heterogeneous and broadly tuned than principal neurons, as previously shown in the ACx and visual cortex (*Mesik et al., 2015*; *Pi et al., 2013*). Indeed, we found that VIP interneurons respond with a low selectivity for drifting grating visual stimuli. There was only a weak positive correlation between the reinforcement-related and the visual stimulus-driven responses. This small correlation could reflect differences in excitability, but more importantly indicates that VIPs can play in both leagues: in region-specific sensory processing and in transmitting global signals to local microcircuits. This further suggests the absence of distinct populations specializing only in global or local processing.

We also observed arousal modulation of VIP interneuron activity in Mtr cortex, SS cortex, and mPFC, consistent with the previous reports in visual and pre-Mtr areas (*Fu et al., 2014*; *Garcia-Junco-Clemente et al., 2017*; *Jackson et al., 2016*; *Reimer et al., 2014*). Arousal states are usually measured by changes in pupil diameter or running speed. One caveat of comparing reinforcement-evoked responses to arousal modulation is that the delivery of water reward and air-puff punishment usually also drives additional changes in arousal, leading to pupil dilation and/or locomotion. However, the relative change in pupil diameter between the small and large dilation groups was larger than the relative change in the corresponding average reinforcement responses indicating that reinforcement-associated change has a large, arousal-independent component (*Figure 3C, D*). Notable, reinforcers also drove VIP activity when additional pupil dilation and/or locomotion was not detectable. Nevertheless, we found that VIP interneuron recruitment by reinforcers was correlated with pupil dilation, similar to previously documented arousal modulation (*Fu et al., 2014*; *Garcia-Junco-Clemente et al., 2017*; *Jackson et al., 2016*; *Reimer et al., 2014*). The late response phase showed a stronger correlation with the pupil size, whereas the initial, transient phase followed the reinforcer delivery more closely (*Figure 3—figure supplement 1E*). These differences might also explain the striking differences in temporal dynamics in simultaneously recorded mPFC and ACx VIP neuron populations.

The arousal modulation was surprisingly muted in ACx during reward and air puff delivery, as previously noted (*Pi et al., 2013*). This observation may relate to the movement-induced suppression of activity in ACx (*Nelson et al., 2013*) in contrast to the running-induced boosting of responses in visual cortex (*Niell and Stryker, 2010*). Alternatively, the lack of arousal modulation may be explained by the arousal-level-dependent, tone-induced network suppression (*Lin et al., 2019*), a mechanism that is similar to lateral inhibition in V1 and controlled by somatostatin interneurons (*Kato et al., 2017*).

High pupil change trials are usually associated with low baseline arousal and thereby with a high level of tone-induced network suppression, and this could reduce the activity of the VIP cells caused by the reinforcement-associated arousal increase itself, and as a consequence, pupil change - VIP cell activity correlation could be diminished. This effect however, only prevailed in Hit and FA, because 1., in Miss and CR high pupil change was not coupled with low baseline pupil that strictly, and 2., the difference of high vs low pupil change was only ~2-fold in Hit and FA, whereas ~14-fold in Miss and CR (considering the peaks).

Additionally, our behavioral paradigm did not encourage mice to run, hence their small movements produced only weak modulation in VIP activity (pupil: 0.31, locomotion: 0.11), measured over a longer temporal interval. Nevertheless, modulation was similar when restricted to the reinforcement time epoch (0–2 s, *Figure 3*). Taken together, these observations lead us to conclude that changes in arousal alone cannot explain the recruitment of VIP interneurons upon reward or punishment.

The circuit basis of the global signal is not yet known, although neuromodulatory systems are prime candidates, in particular, the forebrain cholinergic system. Indeed, central cholinergic neurons convey rapid reinforcement responses to cortex (*Hangya et al., 2015*), and a type of layer 1 inhibitory neuron is activated by punishment through a nicotinic mechanism (*Letzkus et al., 2011*). VIP neurons express fast, ionotropic nicotinic receptors and can be activated by acetylcholine in vitro (*Alitto and Dan, 2012*; *Chen et al., 2015*). Optogenetic cholinergic stimulation can also depolarize the membrane potential of VIP neurons in vivo (*Gasselin et al., 2021*). Cholinergic activity in the neocortex was necessary for learning and modulated VIP cells during locomotor activity (*Ren et al., 2022*), and potentially also responsible for the co-activation in the auditory discrimination task. However, it remains unclear how this putative reinforcer-mediated cholinergic signaling would be ultimately integrated within cortex as multiple inhibitory neuron types other than VIP interneurons are known to respond to acetylcholine as well (*Kuchibhotla et al., 2017*). Another possibility is that the serotonergic system could convey this reinforcement signals (*Cohen et al., 2015*). Indeed, many (but not all) VIP neurons express 5HT3A, the ionotropic serotonin receptor, which could rapidly transduce information via serotonergic neuromodulation (*Tasic et al., 2016*; *Zeisel et al., 2018*).

VIP interneurons have been found to target excitatory neurons as well (*Kullander and Topolnik, 2021*; *Zhou et al., 2017*), and the CCK+ VIP cells (a smaller subpopulation than the interneuron-specific cells) are known to primarily innervate them (*Guet-McCreight et al., 2020*). Nevertheless, reinforcer-induced activation of VIP interneurons, is likely to produce a net disinhibitory effect (*Lee et al., 2013*; *Pi et al., 2013*) and thereby provide gain modulation (*Pi et al., 2013*) through changing the balance of inhibition across the somato-dendritic axis (*Pfeffer et al., 2013*). Such reinforcer-induced disinhibition-mediated increase in cortical gain could provide a circuit-level explanation for the broad recruitment of dendrites observed during reinforcement (*Lacefield et al., 2019*). Indeed, VIP-mediated disinhibition through somatostatin interneurons will preferentially boost dendritic spikes (*Larkum et al., 2009*; *Lavzin et al., 2012*; *Palmer et al., 2014*; *Smith et al., 2013*). This could in turn support the role of VIP neurons in circuit plasticity in visual cortex (*Fu et al., 2015*) and hippocampus (*Donato et al., 2013*). As VIP neuron subtypes can have diversified functional roles in the microcircuit beside disinhibition (*Kullander and Topolnik, 2021*), future research should also focus on these sophisticated circuit operations.

In summary, we identified a cortex-wide or global activation mode for VIP interneurons, separate from their local circuit function specialized for each cortical region. By signaling reinforcement events, the global activation of VIP interneurons early in learning, may form a new information channel for cortical learning and explain how organism-level information about reinforcers regulates local circuit processing and plasticity. VIP interneurons specialize in disinhibition control over principal cells, producing selective amplification of response and gating of dendritic plasticity. This process could enable associating distant, co-active cortical ensembles predicting significant outcomes, whether reward or punishment. In other words, the VIP-mediated feedback signaling could provide the required global learning signal to strengthen functional connectivity across cortical representations. In this way, VIP neurons may transiently dial up the gain on learning, similar to how different learning phases are controlled in deep neural networks in artificial intelligence (*Guerguiev et al., 2017*).

## Materials and methods

**Key resources table**

| Reagent type (species) or resource | Designation | Source or reference | Identifiers | Additional information |
|---|---|---|---|---|
| Recombinant DNA reagent | AAV9.Syn.Flex. GCaMP6f.WPRE.SV40 | Penn Vector Core | Cat# AV-1-PV2819 | |

*Continued on next page*

*Continued*

| Reagent type (species) or resource | Designation | Source or reference | Identifiers | Additional information |
|---|---|---|---|---|
| Biological sample (*Mus musculus*) | *Vip$^{tm.1(cre)Zjh}$*/J, B6.129P2-*Pvalb$^{tm1(cre)Arbr}$*/J, FVB/N-*Tg(Thy1-cre)1Vln/J*, | The Jackson Laboratory | RRID: IMSR_JAX:010908 RRID: IMSR_JAX:017320 RRID: IMSR_JAX:006143 | |
| Software, algorithm | MATLAB | MathWorks | | |
| Software, algorithm | MES | Femtonics | | |

## Animals

All experimental procedures were carried out following the guidelines of the Animal Care and Experimentation Committee of the Institute of Experimental Medicine of the Hungarian Academy of Sciences, and the Cold Spring Harbor Laboratory Institutional Animal Care and Use Committee, in accordance with the Hungarian, EU, and National Institutes of Health regulations (reference number: PEI/001/194-4/2014). We used male and female adult (6–24 week-old) VIP-Cre, PV-Cre, and Thy-1-Cre mice (*Vip$^{tm.1(cre)Zjh}$*/J, B6.129P2-*Pvalb$^{tm1(cre)Arbr}$*/J, FVB/N-*Tg(Thy1-cre)1Vln/J*, The Jackson Laboratory) housed in small groups of 2–4 under controlled temperature and humidity conditions. They were kept on a reverse light cycle, and during the training and the experimental period the water consumption of the VIP-Cre and PV-Cre mice was restricted to 1 ml/day after recovering from surgery. The mice had ad libitum access to food.

## Surgical procedure

Animals were anesthetized using a cocktail of fentanyl, midazolam, and meditomidine (0.05 mg, 5 mg, and 0.5 mg/kg, respectively). Ropivacaine 0.2% was administered subcutaneously over the skull prior to the incision. After removing the skin over the top of the skull, which was then thoroughly cleaned and dried, a round craniotomy was performed using a 3-mm diameter biopsy punch and a dental drill. After the bleeding had been stopped, a double coverslip was implanted over the cranial window and fixed using a mixture of cyanoacrilate glue (Loctite Superbond) and luting cement (3 M ESPE RelyX). Finally, a metal headbar was cemented to the skull using dental cement (C&B Superbond). The 3-mm diameter cranial window was positioned according to two main aspects. We centered the craniotomy on the injection site, except in Mtr and visual areas, where this would have resulted in transecting the sutures, which would have caused larger motion artifacts and severe bleeding from venous sinuses. During the procedure, the mice were head-fixed and laid on a heating pad to maintain stable body temperature. After the operation, the mice were woken up using a mixture containing nexodal, revertor, and flumazenil (1.2 mg, 2.5 mg, and 2.5 mg/kg body weight, respectively), and put on another heating pad where they stayed until recovered enough to be finally put back in their home cages. Post-operative care consisted of a daily intraperitoneal carprofen injection (0.5 mg/ml, 500 µl) for up to 5 days, and subcutaneous Ringer lactate injection (0.1–0.15 ml) to prevent dehydration. The cranial window implantation was usually performed 2 weeks after the virus injection. Injection sites of the 18 dorsal cortical regions from 16 mice were defined on the basis of coordinates from the Allen and Paxinos brain atlases (*Figure 2A*). In the visual cortex, the correct location was further confirmed by recording the responses of the cells to visual stimulation. Post hoc histology was performed in early experiments to ensure our bregma coordinates matched the Paxinos atlas. Each region was then recorded one time per animal.

## Viral injection

Anesthesia and post-operative care were executed as above. A small, approximately 0.5-mm diameter craniotomy was performed with a dental drill. 200–300 nl AAV9.Syn.Flex.GCaMP6f.WPRE.SV40 (Penn Vector Core) was injected using a borosilicate pipette at 350 µm depth into different cortical areas for two-photon imaging. The speed of the injection was 15–20 nl/s, and there was a 10-min

period between the end of the injection and the removal of the pipette to prevent leakage. We injected one to two areas per animal.

## Optical fiber implantation

Animals were anesthetized using isofluorane (1 l/min O$_2$ – 0.8% isoflurane) and placed in a stereotaxic apparatus. A small craniotomy was performed using a dental drill above the left ACx (2.50 mm posterior to the bregma and 4 mm lateral to the midline) and the right mPFC (1.75 mm anterior to bregma and 0.5 mm lateral to midline). 200 nl of AAV9.Syn.Flex.GCaMP6f.WPRE.SV40 (Penn Vector Core) was then injected at a rate of 50 nl/min into the ACx (1.2 mm deep) and in the mPFC (1.5 mm deep). The fiber optic cannulas (400 µm, 0.48NA, Doric lenses) were inserted 0.4 mm above the injection sites for both locations and sealed in place using Metabond, Vitrebond, and dental acrylic. Behavioral training and physiological recordings were started at least 2 weeks after surgery to allow mice to recover and the fiber to clear.

## Data collection using fiber photometry

Fiber photometry data were collected and analyzed using a custom-made photometry setup and Matlab-based software. We used a 470-nm LED source (M470F3, Thorlabs) coupled to an optic fiber (M75L01) and collimation lens (F240FC-A) for GCaMP6f excitation. The 470-nm excitation light was delivered to the cannula implanted on the head of the animal using a second collimation lens (F240FC-A) coupled to a 400 µm, high NA, low autofluorescence optic fiber (FP400URT, custom made, Thorlabs). The emission light was collected using the same optic fiber and directed to a Newport 2151 photoreceiver using a focusing lens (ACL2541U-A, Thorlabs). Excitation (ET470/24 M) and emission (ET525/50) filters, and a dichroic mirror (T495LPXR) were purchased from Chroma Technology. The 470-nm excitation light was amplitude-modulated at a frequency of 211 Hz, with a max power of 40 µW, using an LED driver (LEDD1B) controlled through a National Instrument DAQ (NI USB-6341). The modulated data acquired from the photoreceiver were decoded as in *Lerner et al., 2015* using a custom Matlab function (available at https://github.com/QuentinNeuro/Bpod-FunctionQC, copy archived at swh:1:rev:1ea70f47f0bd5fbf5441abe0dac3dc70ed3c9a8b; *Chevy, 2021*; *Ibrahim et al., 2013*).

## Auditory discrimination task

Mice were kept on a limited access water schedule for behavioral experiments. They had to lick when they heard a go tone (frequency: 5-kHz complex tones for sessions with two-photon recordings, 3-kHz pure tones for sessions with photometry recordings, duration: 0.5 s) to get small water droplets (5 µl) as a reward, and avoid licking after hearing a no-go tone (frequency: 0.5-kHz complex tones for sessions with two-photon recordings, 20-kHz pure tones for sessions with photometry recordings, duration: 0.5 s) which was associated with a 100-ms long mild air puff aimed into the eye. The absence of licking in go trials was not rewarded (Miss trials). If the animal correctly withheld licking to no-go tones (CR), the air puff was omitted. The time interval between tone and reinforcement onset was a function of the reaction time of the animal. In the figures, responses were aligned to reinforcement onset before averaging in FA and Hit, which resulted in a jitter in cue onset time. Reinforcement came 0.5 s after it was triggered, except for during the fiber photometry recordings where reinforcers were delivered upon licking (see below). The intensity of the air puff was set to yield a blink response. In some experiments, we introduced two additional stimuli that were less easy to discriminate (8 kHz for go and 10 kHz for no-go tones). The addition of these cues did not reveal any significant differences in GCaMP6f signals in VIP neurons, therefore these trial types were combined as go and no-go stimuli for further analysis. Licking was detected using a custom-made infrared sensor. Behavioral data were acquired using a Bpod device (Sanworks, LLC), and the tones were generated using a PulsePal device (*Sanders and Kepecs, 2014*) and Logitech speakers. The photometry experiment used the updated version of Bpod that allows the tones to stop immediately when a mouse licks during the tone. Therefore, the duration of tone in the Hit and FA trials varied. Note that when the behavioral task was set up for the two-photon experiment, this function was not available in Bpod, and the tone duration was fixed to 0.5 s for all trial types. In one set of experiments we measured how pupil dilation changed during behavior. A ×4 objective was attached to a CMOS camera (Basler puA 1600–60 µm) to record pupil diameter and eye movements. In another set of experiments, we recorded running speed: mice

were head-fixed over a rotating plastic plate allowing them to run freely. The rotation speed of the dial was recorded by an optical mouse (Urage reaper 3090, Hama) mounted upside down on the lower side of the plate.

## 3D AO microscopy

The improved microscope is designed and constructed based on the previous system reported earlier (see Figure S1 in *Szalay et al., 2016*). Briefly, short pulses were delivered by a femtosecond laser (Mai Tai, Spectra Physics). The coherent backreflection was eliminated by a Faraday isolator (BB9-5I, EOT). Thermal drift errors of optical elements were compensated for by an automatic beam-stabilization unit (BeamStab, Femtonics). The temporal dispersion was compensated for by a motorized four-prism sequence that could be automatically tuned in the 720–1100 nm wavelength range to provide the required large, negative, second- (up to 100,000 $fs^2$) and third-order (up to 45,000 $fs^3$) dispersion compensation (4DBCU, Femtonics). The 4DBCU unit was fine-tuned to provide the best image contrast and SNR at each wavelength in the depth. The first two water-cooled AO deflectors were filled with chirped acoustic waves whose frequencies form two orthogonal electric cylinder lenses (AO z-focusing unit). The second group of AO deflectors, with 15-mm clear optical aperture (Gooch and Housego), did the majority of lateral scanning and also compensated for the longitudinal and lateral drift of the focal spot in cooperation with the first two deflectors according to *equations S1-S70* published earlier (*Reid et al., 2016*). The deflectors are driven in the 55–120 MHz frequency range, therefore the noise emitted does not interfere with the auditory cues, as mice can't hear it. This, in combination with the water cooling of the deflectors makes the AOD-based scanning the quietest technology for in vivo imaging. The two groups of deflectors were coupled together by a telecentric relay system (using two achromat lenses, #47–318, Edmund Optics) which contained a half wave plate (AHWP10M-980, Thorlabs) to set the optimal polarization for maximal diffraction efficiency. There is a one-to-one relationship (a bijection) between non-linear radiofrequency signals and the position, speed, and direction of the moving focus spot (see *Equations S1-S70* and *Supplementary file 1*, Table S1 in *Szalay et al., 2016*). We used these quadratic equations to change the frequency of the sine wave drive to generate multiple 3D drifts from any arbitrary position at any desired speed. In this way, multiple small frames were generated (3D chessboard scanning) around each VIP cell from 10 to 25 lines. Therefore, not only the somatic signal but also the surrounding background information was detected: this enabled the somatic fluorescent signals to be preserved, even during brain movement, for off-line motion artifact compensation. A second telecentric relay system consisting of two achromatic lenses (#47–319, Edmund Optics, G322246525, Linos) focused the diffracted light beams onto the back aperture of the objective. The back-reflected fluorescence signal was separated from the excitation beam by a long-pass dichroic with a cut-on wavelength of 700 nm (700dcrxu, Chroma Technology). Red and green channels were split using a long-pass dichroic at 600 nm (t600lpxr, Chroma Technology). The absorption filters for green and red fluorescence was centered to 520 ± 30 nm and 650 ± 50 nm, respectively (ET520/60 m, ET650/100 m, Chroma Technology). Two extra infrared filters (ET700sp-2p8, Chroma Technology) blocked the back-scattered excitation beam from the GaAsP photomultiplier (H10770PA-40, Hamamatsu). The entire detector assembly was fixed to the objective and moved together during setting the nominal focal plane for the 3D AO imaging to minimize the detection pathway and maximize photon collection efficiency. A ×20 objective (XLUMPlanFI20×/1.0, water immersion, Olympus) with a 1.0 numerical aperture was used.

## Recording sparsely labeled networks in 3D with AO scanning

The main advantage of 3D AO microscopy is that the entire measurement time can be restricted to the required ROIs: this can result in a $10^7$-fold increase in the product of $SNR^2$ and the measurement speed (see Equations S82-S85 in *Szalay et al., 2016*). Therefore with each frame of the chessboard scanning method we only needed to record less than 5% of each VIP cell to preserve the fluorescence information for motion correction (*Szalay et al., 2016*). From these two parameters (and using Equations S82 and S84 from *Szalay et al., 2016*) we can calculate the increase in measurement speed and SNR for 3D chessboard scanning as follows:

$$\left(SNR_{gain}\right)^2 * v_{gain} = \frac{V_{total}}{\sum_{i=1}^{N_{ROI}} V_i} > \frac{1}{1*2} \sim 5000 \tag{S1}$$

where $v_{gain}$ and $SNR_{gain}$ are the relative gains in measurement speed and SNR, respectively, $N_{ROI}$ is the total number of ROIs, $Vi$ is the volume of region number $i$, and $V_{total}$ is the total scanning volume. This means that we can get over 5000-fold increase in $SNR^2$ or in the measurement speed (or even in the product of both) when AO-based 3D ROI scanning is used instead of point-by-point volume scanning.

To calculate this comparison more quantitatively, we compared 3D chessboard scanning with point-by-point scanning, volume scanning, and multi-layer imaging when AO scanning or resonant scanning with fast piezo z-drive were used (see *Supplementary file 1*, Table S1). We limited our comparison to these point scanning methods because they allow whole-field detection and, therefore, deep penetration in vivo. We recorded 120 VIP cells in a 689 µm×639 µm×580 µm volume with 548×507×193 pixel resolution using 3D chessboard scanning (*Figure 1A*). The 3D chessboard scanning method could image the 120 chessboards at 27.7 Hz (*Figure 1A*, *Supplementary file 1*, Table S1). However, the measurement speed was only 0.00062 Hz when the same 120 neurons were recorded using point-by-point volume scanning when using the same, relatively long, pixel dwell time (30 µs). This means a 44,762-fold lower measurement speed. We saw a smaller reduction in measurement speed when we compared chessboard scanning with resonant scanning. The highest speed of the currently available resonant scanners is about 16 kHz, corresponding to ~0.1 µs (=1/16 kHz/548 pixel) pixel dwell time which results in a 0.16 Hz volume-scanning speed (*Supplementary file 1*, Table S1) which is too slow to resolve $Ca^{2+}$ responses. Moreover, as the pixel dwell time is 243-fold lower we would collect less signal from one pixel, resulting in a 41,506-fold decrease in the product of $SNR^2$ and measurement speed (*Supplementary file 1*, Table S1). We could accelerate measurement speed by restricting the numbers of the recorded z-layers to 19 because VIP neurons were present only in 19 z-layers in the exemplified measurement. However, in this case, we also needed to add about 20-ms setting time for each z-layer because the long-range z-drives required higher setting times according to the specifications of the piezo-actuators (see, e.g. https://www.physikinstrumente.com). This resulted in a measurement speed of ~1 Hz and in 6654-fold decrease in the product of $SNR^2$ and measurement speed when compared to 3D chessboard scanning (*Supplementary file 1*, Table S1). The increased SNR of the 3D chessboard scanning allowed the reduction of the laser intensity which resulted in lower phototoxicity according to the LOTOS (low-power temporal oversampling strategy *Chen et al., 2012*). The LOTOS-based multi-photon imaging is one of the main advantages of AO scanning and provides long lasting imaging in chronic behavior experiments.

During these comparisons we did not consider two important technical factors in our calculations. First: the gain in SNR was calculated only for a single pixel (which is a volume element in space, therefore we can name it as voxel). However, both 3D chessboard scanning and volume scanning capture multiple voxels from a single VIP neuron (in our measurements for chessboard scanning: 105.2 ± 0.4 voxels/neuron and for volume scanning: 338.5 ± 0.1 voxels/neuron). Therefore, in a more precise calculation we need to divide the improvement shown in Table S1 for chessboard scanning with the ratio of 338/105.

Second: piezo actuators and resonant scanners are mechanically never perfectly balanced and are also sensitive to local mechanical vibrations and thermal turbulences which result in tumbling, wobbling, and jitter in the laser scans. These mechanical effects are difficult to precisely quantify into fluorescence changes although they would compensate the first factor. Therefore, for simplicity of calculation, both factors were ignored in our calculations.

Random-access targeting of ROIs by AO scanning is useful not only in 3D but also in two-dimension (2D). Because the ratio of the VIP cells in the cortex is about <1% we can estimate the increase in measurement speed and SNR in 2D as follows:

$$\left(SNR_{gain}\right)^2 * v_{gain} = \frac{A_{total}}{\sum_{i=1}^{N_{ROI}} A_i} \sim 100 \tag{S2}$$

## Visual stimulation

A liquid-crystal display (LCD) monitor was placed at a distance of 20 cm from the contralateral eye of the mouse, spanning 100° × 70° of the visual field. The objective was covered with a black rubber shield to prevent stray light entering through the gap between the animal's head and the objective. A visual stimulation protocol was written in Matlab using the 'Psychtoolbox' package. The protocol consisted of eight differentially directed gratings with an angular interval of 45°. At the beginning of each trial, a gray screen was presented for 20 s. After that, a grating appeared and remained still for

1 s, and then moved orthogonally to its orientation for 6 s at 1 cycle/s speed, then it stayed still for 1 s, and finally the gray screen reappeared again. Gratings were repeated 10–20 times per direction in pseudorandom order. Pyramidal cell data were obtained from Thy-1-Cre mice.

## Two-photon imaging data analysis

Motion correction, selection of ROIs corresponding to VIP cells on the frames of 3D chessboard scanning, background calculation, ΔF/F calculation, filtering, and data visualization were performed using the MES data acquisition software written in Matlab and C++ (Femtonics). Motion correction, if necessary, was conducted with a custom-written offline motion correction algorithm (see Off-line motion corrections section), and remaining artifacts were interpolated or smoothed with partial Gauss filtering under visual control.

For the trial-to-trial analysis, we considered a neuron in a given trial as active if the difference between the peak ΔF/F value of reinforcer delivery epoch (0–2 s interval after reinforcement onset) and mean ΔF/F value of baseline epoch (−2–0 s interval before tone onset) was higher than two SD. Peak ΔF/F value was defined here as the average ΔF/F value of the datapoints around the peak in the range of 250 ms. The results presented in *Figure 2—figure supplement 1* used two SD as a cutoff. We defined synchronicity as the number of active neurons divided by the total number of all neurons in a given a trial, i.e., how many neurons are activated simultaneously. Reliability was calculated as the number of active trials divided by the total number of trials, i.e., how reliably the neuron is activated in Hit and FA trials. Stability means the percentage of active neurons across trials and sessions.

We used linear regression models to address heterogeneity of the cue responses (*Figure 2A*, *Figure 2—figure supplement 2D*). The explanatory variable for the first model was the hit rate (number of Hit trials divided by the number of go trials) to characterize behavior. The dependent variable was the relative size of the average cue response compared to the average reinforcement response as a reference. The values were calculated on the population average traces of the Hit trials from each measurement. In the second model we used categorical variables with dummy coding for four functional regions (Mtr, sensory, parietal, visual) as explanatory variables to describe regional differences. The regression was fitted using ordinary least squares method of Statsmodels package in Python 3 based Anaconda data science platform.

The clustering analysis (*Figure 2F–G*, *Figure 2—figure supplement 2A, B*) was done using a custom Matlab routine. Positive somatic $Ca^{2+}$ responses recorded during Hit trials were extracted. 16 cells were excluded from this analysis as they disproportionately increased the reconstruction error. Data were z-scored using the mean and variance of fluorescence during the first second of recording. We further normalized using the maximal amplitude of the response calculated during the period from 0 to 4 s after reward delivery. We applied a dimensionality reduction along the time axis using a PCA on the period from 0 to 4 s after reward delivery. We then considered only the first four principal components (PCs) explaining 90% of the data for clustering purposes. K-means clustering with five replicates was used to cluster into five types the PCs of the responses of VIP interneurons.

We estimated cell diameters by recording the fluorescence profile of the somata placing two lines along the longest and the shortest diameters (*Figure 2—figure supplement 3A*). Then the averaged and Gauss-filtered fluorescence profiles were used to calculate FWHM after subtracting the fluorescent baseline and normalization.

We applied TCA (*Kolda and Bader, 2009*; *Williams et al., 2018*) on somatic $Ca^{2+}$ responses recorded during the discrimination task. After smoothing, single-trial neural activities corresponding to the reaction time periods for Hit and FA trials were time-warped to a fixed 1.5 s/30 data points in length. This step was necessary because (1) the reaction time varied from trial to trial; and (2) the variable number of neurons recorded in the different session led to a variable sampling rate for imaging. These factors led to a trial- and session-dependent number of images between the cue and the reward delivery. All recordings were rendered non-negative by subtracting the minimal fluorescence ΔF/F value for each cell. Data were finally normalized by dividing by the average maximum fluorescence ΔF/F value of Hit-only trials. Only the first 10 trials of each types were then selected for each cell and each session and assembled in a N×T×K matrix where N=771 neurons, T: time (s), K=40 trials. TCA reconstruction error was computed with different latent numbers [1,2,3,4,5,10,15,20] with 10 different initial conditions for each latent number. Using three latents led to a reconstruction error of 22%.

A generalized linear model was built to determine the contributions of the behavioral events to the VIP interneuronal activity (*Figure 3—figure supplement 2D*). This model used auditory cue, locomotion velocity, pupil diameter, water, and air puff delivery as explanatory variables. Estimated amount of water accessible at the lick port was represented with a kernel consisting of a Gaussian fit that started at water delivery and peaked when the average licking activity of the three mice reached a top before a plateau phase. Then the kernel decayed in a linear fashion until it reached the baseline at a time point that was calculated to match the end of the plateau of licking activity (5% decrease from peak). Air puff was incorporated as a Gaussian kernel started at the onset of the air puff delivery and peaked at 0.2 s that matched the time of the valve closure. The trace representing pupil diameter was Gauss filtered. Variables were downsampled to match the frame rate of the calcium recordings. All predictors were standardized in order to have a fair penalization in the subsequent regularized regression. Then a lasso regression was fitted with automated hyperparameter tuning and fivefold cross-validation. This type of regression can handle data with multicollinearity and reduces overfitting. The penalty factor decreased the weights of the unimportant explanatory variables. Adding an explanatory variable containing random floating point numbers in the interval [0,1] did not change the predictive power of the model, and this variable did not contribute significantly to the prediction.

## Pupil diameter

The video recorded with the camera was first thresholded to isolate the pupil on the image. The pupil area was fitted to an ellipse, and the main diagonal was extracted. Missing frames caused by spontaneous or air puff-triggered blinking were interpolated manually. A Gaussian filter was applied to smooth eye movement-related artifacts. In analyzing the change of pupil diameter, the traces were normalized to $\Delta P/P = (P(t) - P_0)/P_0$ using a 2-s period before tone onset as baseline ($P_0$). Trials of each outcome were separated to high- and low-arousal groups on the basis of the change in the pupil diameter. The area under the pupil diameter curve was calculated in the 0–3 s interval after reinforcement onset, and the median value was selected. If the area under the curve value of a given trial was higher or lower than the median value, it was considered to be high- or low-arousal trial, respectively. For baseline pupil analysis, when we compared the VIP cell activity after reinforcement associated with low and high baseline arousal levels, the trials were again separated into low- and high-arousal trials, but here, the basis of the separation was the area under the pupil diameter curves in the [−2;0] s interval before the tone onset. In *Figure 3—figure supplement 2A, B*, we separated a subset of trials in which reinforcement was not accompanied by pupil diameter and locomotion velocity increase or it did not step over two SD. Pupil diameter, velocity, and fluorescence calculations were made on the [1;0] s interval after the reinforcement onset in this analysis.

## Locomotion velocity analysis

Velocity traces were first Gauss filtered. They show the absolute speed of the movement, regardless of the direction. In Hit and FA trials, we defined the change in the running speed as the speed difference between the reinforcer delivery time period (0–2 s interval after reinforcement onset) and baseline time period (−2–0 s interval before tone onset). In Miss and CR trials, the speed difference was calculated between the tone delivery time period (0–2 sec interval after tone onset) and baseline time period (−2–0 s interval before tone onset). Trials were separated into low and high speed change groups according to the median speed change value.

## Visual stimulation

OSI and DSI were calculated as $OSI = (R_{pref} - R_{ortho})/(R_{pref} + R_{ortho})$, and $DSI = (R_{pref} - R_{opp})/(R_{pref} + R_{opp})$ (*Schumacher et al., 2019*), where $R_{pref}$ denotes the amplitude of the response to the preferred orientation (OSI) or direction (DSI), $R_{ortho}$ denotes the response to the orthogonal orientation in the OSI formula, and $R_{opp}$ refers to the response to the direction that is opposite to the preferred one.

## Statistical analysis

For all analyses, the activation/suppression period was set to be 0–2 s after stimulus onset: the statistical significance of the change of $Ca^{2+}$ responses was then evaluated and compared to a stimulus-free baseline (−2–0 s before stimulus onset). The statistical significance of the activation and suppression was determined by a p-value cutoff of 0.05. First, the mean baseline values of each

trial were subtracted from each $Ca^{2+}$ trace in order to directly compare the effect of stimuli on $Ca^{2+}$ responses and minimize the effect of unknown sources of noise. Lilliefors normality test was used to evaluate whether the $Ca^{2+}$ signals of individual VIP neurons followed a normal distribution. The Lilliefors test showed that 76% of VIP neurons (Hit: 70%, FA: 82%) followed a normal distribution. The fraction of the neurons activated by reinforcers (see below) with normal distribution (91%) was similar to that of activated neurons with non-normal distribution. Therefore, we used a one-tailed one sample t-test to classify the activation and suppression (see Supplemental information for a more sensitive analytical method). Neurons were classified as responsive when either activation or suppression was statistically significant. Student's t-test (*p<0.05, **p<0.01, ***p<0.001) was also used to compare calcium responses associated with low and high arousal, or low and high running speed. PCA loadings of different areas and TCA trial factors of trial types with and without reinforcement were compared with Mann-Whitney test. If not otherwise indicated, data are presented as mean ± SEM.

## Off-line motion correction

In the case of chessboard scanning, neuronal somata were selected from a z-stack, then the selected square ROIs were arranged as a 2D chessboard. Since the motion of a single frame during the scanning period as well as the relative rotation between subsequent frames, were not relevant, the transformation to be corrected could be approximated by a simple translational transformation between the scanning periods of the different frames. For efficiency, an algorithm based on fast Fourier transformation was used (*Fuster and Bressler, 2015*; *Guizar-Sicairos et al., 2008*). The template images were chosen either manually or by selecting the best correlating 20% of the relevant ROIs on all frames.

In some cases, images were also preprocessed, by either adaptive histogram equalization or simple median filtering. The image registration algorithm also provided the error of matching the moving images to the template images. As the drifting and scanning parameters were identical for each scanned ROI, we calculated the final displacement vector as the median of a fixed percentage of all ROIs with the smallest matching errors.

## Code availability

Custom written analysis codes are available at https://github.com/QuentinNeuro, (copy archived at swh:1:rev:1ea70f47f0bd5fbf5441abe0dac3dc70ed3c9a8b; *Chevy, 2021*).

## Acknowledgements

We thank Lídia Popara, Áron Szepesi, and Alexandra Bojdán for technical help, and all members of the Kepecs and Rózsa labs for their helpful comments. This study was supported by KFI-2018–00097, VKE-2018–00032, NKP-2017–00001, KTIA_NAP_12-2-2015-0006, 2017–1.2.1-NKP-2017–00002, GINOP_2.1.1-15-2016-00979, and János Bolyai Research Scholarship of the Hungarian Academy of Sciences. The project was implemented with the support from the National Research, Development and Innovation Fund of Hungary financed under the VKSZ_14-1-2015-0155, KFI_16-1-2016-0177, NVKP_16-1-2016-0043 funding scheme. This project received funding from the European Research Council (ERC) under the European Union's Horizon 2020 research and innovation programme (grant agreement No. 682426 and VISGEN_734862, 712821-NEURAM). HP is supported by NARSAD Young Investigator Grant and NIH R01MH110391. The study received funding from the National Institutes of Health, R01NS075531 and R01NS088661, to AK.

## Additional information

### Competing interests

Gergely Katona: is a founder of Femtonics Ltd. Balázs Rózsa: is a founder of Femtonics Ltd. and a member of its scientific advisory board. The other authors declare that no competing interests exist.

## Funding

| Funder | Grant reference number | Author |
| --- | --- | --- |
| European Research Council | | Gergely Szalay |
| National Institutes of Health | R01NS075531 | Adam Kepecs<br>Hyun-Jae Pi |
| National Institutes of Health | R01MH110391 | Adam Kepecs<br>Hyun-Jae Pi |
| National Institutes of Health | R01NS088661 | Adam Kepecs<br>Hyun-Jae Pi |
| Horizon 2020 | 682426 and VISGEN_734862 | Gergely Katona |
| Horizon 2020 | 712821-NEURAM | Gergely Katona |
| National Research, Development and Innovation Fund of Hungary | VKSZ_14-1-2015-0155 | Gergely Katona |
| National Research, Development and Innovation Fund of Hungary | KFI_16-1-2016-0177 | Gergely Katona |
| National Research, Development and Innovation Fund of Hungary | NVKP_16-1-2016-0043 | Gergely Katona |
| Hungarian Academy of Sciences | János Bolyai Research Scholarship of the Hungarian Academy of Sciences | Balázs Chiovini |
| NARSAD | Young Investigator Grant | Hyun-Jae Pi |
| NKFI Hivatal | 2020-2.1.1-ED-2021-00190 | Balázs Rózsa<br>Zoltán Szadai<br>Katalin Ócsai |
| NKFI Hivatal | 2020-2.1.1-ED-2022-00208 | Balázs Rózsa<br>Zoltán Szadai<br>Katalin Ócsai |

The funders had no role in study design, data collection and interpretation, or the decision to submit the work for publication.

## Author contributions

Zoltán Szadai, Data curation, Formal analysis, Investigation, Writing – original draft, Writing – review and editing, Visualization; Hyun-Jae Pi, Conceptualization, Resources, Formal analysis, Investigation, Visualization, Writing – original draft, Writing – review and editing; Quentin Chevy, Data curation, Software, Formal analysis, Validation, Investigation, Visualization, Writing – original draft, Writing – review and editing; Katalin Ócsai, Software, Formal analysis, Methodology, Writing – original draft, Writing – review and editing; Dinu F Albeanu, Resources, Validation; Balázs Chiovini, Validation, Investigation; Gergely Szalay, Resources, Validation, Investigation; Gergely Katona, Resources, Software, Formal analysis, Supervision, Funding acquisition, Visualization, Writing – original draft, Project administration, Writing – review and editing; Adam Kepecs, Conceptualization, Resources, Software, Supervision, Funding acquisition, Investigation, Visualization, Writing – original draft, Writing – review and editing; Balázs Rózsa, Conceptualization, Resources, Formal analysis, Supervision, Funding acquisition, Investigation, Visualization, Writing – original draft, Project administration, Writing – review and editing

## Author ORCIDs

Zoltán Szadai http://orcid.org/0000-0002-6286-4686
Balázs Rózsa http://orcid.org/0000-0003-1427-7003

## Ethics

All experimental procedures were carried out following the guidelines of the Animal Care and Experimentation Committee of the Institute of Experimental Medicine of the Hungarian Academy of Sciences, and the Cold Spring Harbor Laboratory Institutional Animal Care and Use Committee, in accordance with the Hungarian, EU, and National Institutes of Health regulations (reference numbers: PEI/001/194-4/2014).

## Decision letter and Author response

Decision letter https://doi.org/10.7554/eLife.78815.sa1
Author response https://doi.org/10.7554/eLife.78815.sa2

## Additional files

### Supplementary files

• Supplementary file 1. Comparison of different scanning methods. Scanning speed was calculated according to the equations in the column 'calculation of scanning speed'. Ratio of collected photons was calculated from relative pixel dwell times. All parameters used for calculations are listed in the bottom field. Note, that chessboard scanning provides 170-fold faster measurement speed and 244-fold higher photon collection compared to volume scanning with resonant mirrors.

• Supplementary file 2. Calculation of ratio of responsive neurons following reward.

• MDAR checklist

### Data availability

Imaging data have been deposited at Open Science Framework (project name: CortexWide_VIP).

The following dataset was generated:

| Author(s) | Year | Dataset title | Dataset URL | Database and Identifier |
|---|---|---|---|---|
| Szadai Z, Pi HJ, Chevy Q | 2022 | CortexWide_VIP | https://doi.org/10.17605/OSF.IO/9FBG7 | Open Science Framework, 10.17605/OSF.IO/9FBG7 |

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

## Appendix 1

### Baseline arousal level modulates VIP activity

We split the trials based on the baseline or inter-trial arousal level (*Figure 3—figure supplement 1B*). Thus, trials were split by the median value of the baseline pupil diameter. This analysis revealed that the arousal level in the baseline period inversely correlated with the degree of vasoactive intestinal polypeptide (VIP) activation by reinforcers. For instance, when baseline arousal level was low, the reinforcers tended to induce a stronger increase in VIP activity. When baseline arousal level was high, the reinforcers induced a smaller increase (high vs low baseline: somatosensory: 30 vs 39% ($\Delta$F/F), n=26, p<0.01; motor: 23 vs 30% ($\Delta$F/F), n=111, p<0.001; *Figure 3—figure supplement 1B*). Interestingly, the anti-correlation was also observed between baseline pupil diameter and the increase in pupil diameter. When the baseline pupil diameter was small, the increase in pupil diameter tended to be higher and vice versa (*Figure 3—figure supplement 1B*). Taken together, these results indicate that the baseline arousal level is another important factor that modulates VIP activity.

### Identifying responsive neurons

In this section, we will introduce a new method for selecting responsive neurons from large neuronal populations recorded simultaneously. The method is based on the following: (1) a thresholding method in which neurons with very poor signal-to-noise ratio (SNR) are eliminated at the beginning of the analysis; (2) baseline calculation in the pre-stimulus period; and (3) one sample t-test in the response period.

### 1) The thresholding method

Majority of the cells showed robust spontaneous and reinforcement-related responses with variable amplitude and frequency. However, in some neurons, responses were below the detection threshold in a recording period of over 5 min. To eliminate neurons with low SNR, we calculated the mean amplitude of the 15 ± 5% largest peaks detected in $Ca^{2+}$ transients during the recording period of over 5 min and divided it by the average, pre-stimulus STD. Neurons with a (mean amplitude)/SD ratio below 5 were eliminated from the analysis. This threshold eliminated 4.98 ± 0.01% of the VIP cells (*Supplementary file 2*, Table S2).

### 2) Baseline calculation

There are many ways to define a neuron as responsive or non-responsive. For all definitions, we need a baseline relative to which responsiveness can be calculated. The simplest approach is to define a pre-stimulus temporal interval ($[T_{01}, T_{02}]$) before the cue onset as a baseline period, and calculate the mean, $\mu_0$ (see below).

### 3) One-sample t-test

In a one sample t-test, the null hypothesis is that the population mean is equal to a specified value ($\mu_0$).

$$t = \frac{\bar{x} - \mu_0}{\frac{SD}{\sqrt{n}}} \tag{S3}$$

where SD and n are the standard deviation and sample size, respectively. The degrees of freedom (DF) is n−1. The distribution of the population of sample means x is assumed to be normal although this is not required for the parent population. The distribution of $t_p$ will be approximately normal N(0,1) according to the central limit theorem. If we substitute SD with $\sqrt{n} \times SEM$ where SEM is the standard error of the mean, we get the following criterium:

$$t \times SEM = \bar{x} - \mu_0, \tag{S4}$$

In Student's t-test the $x_0$ hypothesis is accepted as significant if $|t| > t_p$ where $t_p$ is defined as $P\left(|t| > t_p\right) = p$, where P denotes probability. The Student's distribution defines $t_p$ at a given DF *(n−1)* and a given *p* value. Therefore, we can define the following criterium for significance:

$$t_p \times \text{SEM} < \bar{x} - \mu_0 \ or \ t_p \times \text{SEM} < -(\bar{x} - \mu_0) \tag{S5}$$

The simplest approach to define a neuron as responsive is to calculate in a one-sample t-test whether the mean response ($x$ of the neuron is significantly larger [or smaller] in a given interval after 0 ms

(where 0 ms is the time of the stimulus) than the baseline average value (0, which is equal to zero by definition). According to this definition and *Equation S5*, a neuron is responsive if its average time-dependent response transient, $x(time)$ is larger during a given time interval than the product of $t_p$ and the SEM of the population:

$$t_p \times \text{SEM} < \bar{x}(\text{time})|_{[T1,T2]} \tag{S6}$$

and for significantly smaller responses (for inhibition) we can use the modified second Equation from *Equation S5*:

$$t_p \times sem < \bar{x}(time)|_{[T1,T2]} \tag{S7}$$

In practice, we defined the pre-stimulus baseline period, from 2 s before cue onset to the cue onset time. The interval of responses was defined from the time of the reinforcement time (0 s) to 2 s after the reinforcement. This also means that both $T_1$ and $T_2$ time values must be part of the [0 s, 2 s] response interval. In theory, there is no limit to the minimum length of the [$T_1$, $T_2$] interval; however, in practice we used the $T_2 - T_1 \geq 500$ ms criterium which was in the range of the mean length of the single AP potential-induced response at half maximum.

The number of reward and punishment transients collected in a given experiment was variable (between 30 and 50) resulting in variable $t_p$ values which had to be calculated for each experiment separately. For example, a trial with 34 transients means 33 DF (34−1) and $t_p = 1.692$ (p<0.05; two tails). We performed fast 3D recording of VIP interneurons (34–40 cells, from 4 mice, *Supplementary file 2*, Table S2) and calculated the activation ratio. The activation ratio for reward was 95.16 ± 2.09%, which is a much higher ratio than that determined using the standard one sample t-test above (76.22 ± 9.65%).

