## [Editor Report]

The exceptional imaging technique used permitted to detection of the activity of a specific group of cortical neurons known as vasoactive intestinal polypeptide (VIP)-expressing interneurons from several cortical regions with high temporal resolution. The landmark message conveyed by this manuscript is that many VIP-expressing interneurons respond to reward and punishment but also show regional differences. The conclusions drawn are generally supported by the data. This paper is of potential interest to neuroscientists expert in cortical circuitry and behavioral role of neuron types.

---

## [Decision Letter]

**Decision letter after peer review:**

Thank you for submitting your article "Cortex-wide fast activation of VIP-expressing inhibitory neurons by reward and punishment" for consideration by *eLife*. Your article has been reviewed by 3 peer reviewers, and the evaluation has been overseen by a Reviewing Editor and Tirin Moore as the Senior Editor. The following individual involved in review of your submission has agreed to reveal their identity: Francesco Ferraguti (Reviewer #2).

Essential revisions:

1) The submitted data are potentially interesting but preliminary as the activity of VIP+ interneurons is analyzed only in one animal. In order to make the data less anecdotical, data from more animals need to be shown.

2) A revised version of the manuscript should be able to distinguish between reinforcement and arousal, since the reinforcements used in this study may cause arousal

3) The manuscript should assess whether well-trained animals display large responsiveness of VIP neurons to reward or reduced responses.

4) The authors should revise their interpretation of the data based on the current view of cortical VIP+ cells as a diverse population of neurons that mediate both inhibition and dis-inhibition.

5) Data analysis and representation should improve according to reviewers comments – see for example comments of reviewer 3 to submitted figures.

*Reviewer #1 (Recommendations for the authors):*

1. The authors need to consider the current view of cortical VIP+ cells as of providers of both inhibition AND disinhibition (Zhou et al., 2017, Cerebral Cortex). In fact, VIP+ cells are highly heterogeneous, and their diversity and functional role across different cortical areas have been highlighted (Guet-McCreight et al., 2020; Kullander and Topolnik, 2021). This is especially important given functional heterogeneity in the response profiles of VIP+ cells observed in the current study across cortical regions. With the help of PCA and k-means clustering method, recorded VIP interneurons were categorized into 5 groups based on their temporal dynamics. These cells possibly belong to different subtypes of VIP interneurons. Post-hoc identification of these neurons using IHC would possibly provide more information on whether the 5 groups of neurons belong to distinct subtypes of VIP cells and ultimately if there are any differences in the distribution of VIP subtypes across depth.

2. Was there a small fraction of VIP+ cells activated by all tasks (Hit, Miss, CR and FA; Figure 1C)? Not sure, why the AVG response/area is shown. As different cell types were sampled, the best would be to arrange them by depth/layer using the full advantage of the fast volumetric imaging (as shown in Figure S2). Also, cells could be arranged by their soma size given that CR+/VIP+ have a smaller soma when compared to VIP basket cells (Luo et al. 2020).

3. TCA allowed authors to understand the variability in the activity dynamics due to inter-trial variability. The interpretation of the obtained results should be added to the main text (Figure 3H) for better understanding.

4. VIP interneurons from all regions under investigation but the auditory cortex showed arousal and locomotion modulation of reinforcement-mediated recruitment of VIP interneurons. The possible reasons need to be discussed in the main text. Similarly, the absence of arousal (pupil diameter change based) modulation of reinforcement response for Hit & FA trials (unlike Miss & CR trials) requires explanation.

5. Fiber photometry recordings from the auditory cortex (Figure 2A) reveal higher calcium transients for Hit and FA compared to CR and Miss trials, however, the data from Figure S4A shows higher ΔF/F (%) for Miss & CR compared to Hit & FA. Whether the results in Figure 2A and Figure S4A were obtained from different sets of mice should be mentioned.

6. Authors conclude in the Discussion section that – only the late response phase showed a correlation with the pupil size, whereas the initial, transient phase followed more closely the reinforcer delivery, however in the Figure 3 D-E, ΔF/F (%) for Hit & FA from high vs low pupil size change trials in the late response phase (~2-3 sec from the outcome) show roughly similar responses (especially in motor cortex Figure 3D right). The clarification of how the initial and late response phase was defined is necessary for better understanding. A graph comparing the responses (ΔF/F) of VIP interneurons for high vs low pupil size change trials in initial vs late phase would clarify the extent of the influence of arousal on reinforcement-mediated recruitment of VIP interneurons. Moreover, comparison of ΔP/P in Figure 3C and Figure S4A reveals that reinforcement stimuli trials (FA & Hit) lead to higher pupil size change compared to Miss & CR trials. Considering the modulation of VIP recruitment due to arousal (pupil size change), the reinforcement-mediated recruitment of VIP interneurons might be strongly influenced by arousal. More clarification is needed to conclude about reinforcement-mediated VIP recruitment.

7. In the Discussion section (line 440-441), the authors claim that the VIP activity modulation due to locomotion is weak, but the comparison of activity change due to arousal (Figure 3 C-E) and locomotion (Figure 3 F-H) shows that the ΔF/F (%) values in all brain regions for arousal and locomotion modulation are comparable. The claim of weak locomotion modulation of VIP activity is not supported by the results. Also, speed modulation was assessed at a very low animal speed (avg 1.2 cm/s, which was used as a high speed). Have authors tried to look at higher speed locomotion, which has been used in the previous studies?

8. The authors need to discuss the potential mechanisms of co-activation of VIP+ cells (lines 169-173).

*Reviewer #2 (Recommendations for the authors):*

– At least for one area (e.g. MPta) the interindividual variance of VIP+ interneuron activity to the different trials and responses has to be provided. This would importantly add to the understanding of the sources of heterogeneity in the recruitment of VIP+ interneurons.

– The stability of the VIP+ interneuron response to Hit and FA across trials should be shown.

*Reviewer #3 (Recommendations for the authors):*

A discrepancy exists with published data that do not show large responsiveness of VIP neurons to reward in visual and barrel cortices. The authors hypothesize that VIP neurons might only respond to reward and punishment during learning. This is the reason they use mice in the phase of task acquisition, that still have "enough" FA trials, while the published studies measure VIP activity in overtrained mice. To solidify this suspicion, the authors could show that, once their mice have fully mastered the task, the reinforcer responses in VIP cells are in fact reduced or absent. This could harmonize their results with these published studies and significantly strengthen the paper.

Figure 1: Why are the averages in Figure 1 C and D not corresponding? From the legend it appears all curves should be the averages of the above shown 120 VIP cells. Yet their relative amplitude, as well as time course is definitely different. The hit-curve in 1C seems to start to rise before reward delivery, yet this does not seem to be the case in 1D. In 1D it is unclear what the CR and miss trials are aligned to. Is there really error bars on the bottom plots in 1C as mentioned in the legend?

Was MPta chosen for this figure simply because it yielded the biggest n? In Figure 2 it appears to almost be an outlier of all the imaged areas, as it is one of very few areas where VIP neurons appear to entirely lack sensory responses.

Figure 2: In panel A it is exceedingly hard to distinguish the thin lines from the same color thick lines and thus see potential sensory responses in any of the areas, perhaps one could change the hue slightly or use the different shade of blue as in panel H? In general, a different color scheme could be considered for accessibility reasons. And a legend should be added.

For the speaker sign at the bottom of B it is unclear where the start of the auditory cue is, and if it is the same for all the lines of a plot.

The time courses here, as well as the ones in F make it seem as though in this data set there was a fixed interval between cue presentation and reinforcer delivery. Or is it similar to π et al., where the animals triggered the reinforcer by licking. This is not mentioned anywhere.

One session in M1 (hits), as well as one session in M2 (FAs), VIP neurons seem to respond before the onset of the auditory stimulus (judging from the other areas time courses), why is this?

All abbreviations should be explained in the legend (esp. panels F,G,H)

Figure 3 C,D,E: why were these areas chosen, or which areas are included in each of these plots? Do differences between hits and FAs exist? There appear to be potentially interesting areal differences.

In the figures of the time courses of the ACx repsonses (2A and S1D) it seems like VIP cells only have sensory responses in miss and CR trials. Why is this bump absent from hit and FA trials?

line 227: How are these 606 reward delivery sensitive cell related to the 622 in Figure 2?

For the arousal and running modulation: why is the data split here and no correlation calculated as for the relationships to behavioral performance and sensory tuning?

The running paragraph: it needs to be clarified whether they look at running speed per se, as much of the paragraph is written (fast and slow running conditions), or at speed changes, as the beginning of the paragraph states and the values (partly negative) would make it seem. If it is indeed speed change, they should rename the conditions.

Figure S2F seems very important. Was also uncued negative reinforcement presented? How does the uncued data compare to the cued?

Is S5A the exact same data as in 4D, just plotted in a different way? A statistical test should be performed to judge the significance of this correlation.

The methods need to be improved. As is there is not enough detail and analyses cannot be understood properly. For example: What does smoothing according to the reaction time periods mean? Reaction time periods are not even mentioned in the description of the task. Please explain time-wrapping. Why is 1.5s 30 data points when the sampling rate is 27.7 Hz and not 20Hz? Why are only 10 trials selected? Why the first? Why is K then 40?

Supplementary figure S3C is never mentioned.

---

## [Author Response]

Essential revisions:1) The submitted data are potentially interesting but preliminary as the activity of VIP+ interneurons is analyzed only in one animal. In order to make the data less anecdotical, data from more animals need to be shown.

We thank the reviewer and the editor for pointing out this potential misunderstanding in the method description. The AOD data presented in this manuscript are actually from 16 animals. Each animal has been recorded in one (14 mice) or two areas (2 mice). We have 4 recordings from the primary somatosensory cortex, 6 recordings from the primary and secondary motor cortices, 4 from the lateral and medial parietal cortices, and 3 from the primary visual cortex. Fiber photometry has been performed in a different 6 animals hence totaling 22 mice. In addition, our measurements were robust at the level of individual sessions: fluorescence signals were recorded in a high number of trials from multiple neurons. More specifically, Figure 1 shows recording of 128 trials from 120 neurons resulting in a total of n = 120 × 128 = 15,360 transients. Our main data set (e.g., Figure 2) is based on 811 cells × 95-195 trials ≈121,650 transients. Across all these animals and brain areas, we observed a consistent recruitment of VIP interneurons by reward and punishment. Although we reported the number of mice and sessions the recordings were obtained from, to clarify this misunderstanding we have now repeated this several times at the beginning of the Results section and in the figure legends to highlight the robustness of our conclusions.

2) A revised version of the manuscript should be able to distinguish between reinforcement and arousal, since the reinforcements used in this study may cause arousal

Thank you for raising this important point as it is indeed an important confound for the study of reinforcement-related signals. We had addressed this question in Figure 3A-E and related Figure 3—figure supplement 1A-C. Briefly, we showed that while arousal (as measured by pupil dilation) or locomotion influence the activity of VIP interneurons, the reinforcement response from VIP interneurons is maintained independently of the arousal or locomotion states of the animals during the different epochs of the task (baseline or reinforcement delivery period). To further confirm our findings, we are now adding a new supplementary figure (Figure 3—figure supplement 2) showing a subset of trials in which the animals were in high arousal states (i.e., large pupil typically also associated with locomotor activity) prior to the start of the trial, and for which the reinforcement delivery did not trigger any additional elevation in arousal. For these trials, the VIP interneuron activity still increased upon reward delivery.

In addition, we also present a generalized linear model (Figure 3—figure supplement 2D) to quantitatively estimate the contributions of arousal, locomotion, and reinforcement on the VIP neuronal activity in a systematic way. The model showed that arousal does indeed contribute to the variance in VIP signal as previously reported. However, this systematic analysis also highlighted our new finding that VIP-IN activity reports reinforcement delivery. We hope that this can further strengthen our demonstration that reinforcement-related activity of VIP cells is not just a reflection of fluctuations in locomotion and arousal.

3) The manuscript should assess whether well-trained animals display large responsiveness of VIP neurons to reward or reduced responses.

We agree with the editor and the reviewers on this important point. Many studies have started to address directly or indirectly the effect of learning on reinforcer-related responses of VIP interneurons (Turi et al., 2019, Krabbe et al., 2019, Ren et al., 2022). However, the overall goal of this paper is to focus on primary reinforcers (reward and punishment) to demonstrate that they globally recruit VIP interneurons. Now we emphasize this also in the abstract. Nevertheless, it was already evident that during this process the amplitude of the cue-associated responses are proportional with the hit rate: that was the reason why we added Figure 2—figure supplement 2D, which shows that the cue-associated component dominates over the reward-associated component in better-trained mice.

4) The authors should revise their interpretation of the data based on the current view of cortical VIP+ cells as a diverse population of neurons that mediate both inhibition and dis-inhibition.

We thank the editor and the reviewers for drawing our attention to this important topic. According to the question of Reviewer #1, we analyzed the distribution of the diameter of VIP somata. We have found that both at the level of the entire population and individual mice neuronal somata formed two separate distributions (Figure 2—figure supplement 3), possibly indicating two populations (CR+ and CR−) of VIP cells shown in a previous study (Luo et al., 2020).

As we did not find any significant differences in the cell diameter of neurons which were responsive or non-responsive to reinforcement (Figure 2—figure supplement 3B), we concluded that different subtypes of VIP interneurons are similarly activated by reinforcers, but the response variability characterized by the PCA (Figure 2—figure supplement 3C) can indeed reflect subtype-specific differences. We have strengthened the discussion about the complex nature of the VIP cells concerning inhibition and disinhibition in the local circuitry.

5) Data analysis and representation should improve according to reviewers comments – see for example comments of reviewer 3 to submitted figures.

We thank the reviewers and the editor for raising these imprecisions in the description of some of the methods, especially the TCA. We have improved the methods and the figures according to the reviewers’ comments.

Reviewer #1 (Recommendations for the authors):1. The authors need to consider the current view of cortical VIP+ cells as of providers of both inhibition AND disinhibition (Zhou et al., 2017, Cerebral Cortex). In fact, VIP+ cells are highly heterogeneous, and their diversity and functional role across different cortical areas have been highlighted (Guet-McCreight et al., 2020; Kullander and Topolnik, 2021). This is especially important given functional heterogeneity in the response profiles of VIP+ cells observed in the current study across cortical regions. With the help of PCA and k-means clustering method, recorded VIP interneurons were categorized into 5 groups based on their temporal dynamics. These cells possibly belong to different subtypes of VIP interneurons. Post-hoc identification of these neurons using IHC would possibly provide more information on whether the 5 groups of neurons belong to distinct subtypes of VIP cells and ultimately if there are any differences in the distribution of VIP subtypes across depth.

This is a very good comment and an extremely important point. As pointed out by the reviewer, the characterization of VIP interneuron subtypes is a very active field with on-going identification of functional, connectivity, and genetic diversity. Although we are not able to perform the IHC the reviewer suggested, we have tried to address the subtype diversity using soma size as a proxy (Luo et al. 2020). In Figure 2—figure supplement 3, we found that VIP cells can indeed be separated into two groups on the basis of cell diameter: however, these groups did not differ in their responsiveness to reinforcers (see also Essential revisions #4). But the PCA clusters indeed showed significant variability: VIP cells with larger diameters were more likely to belong to fast and sustained cluster profiles. One important observation from the cluster analysis is that virtually all clusters show a fast response to reward, but were instead coined mostly based on their late response. These differences in late responses could represent VIP subtype differences with different receptor subtypes, or different integration into the local cortical microcircuit. We have added this new information to the Results section and strengthened the discussion regarding the VIP interneuron subtypes; we also mention the inhibitory effect of the VIP cells.

We now say in the discussion:

“… An analysis performed to address cell-type diversity using diameter as a proxy separated the cells into two populations, maybe reflecting the CR+/− classification (Luo et al. 2020). Given the high proportion of VIP neurons responding to reward and punishment, it seems likely that multiple subtypes of VIP interneurons respond to reinforcers. A potential subtype--heterogeneity manifested itself in the late response phase. Further studies using inter-sectional targeting strategies will be required to provide insight for the potential cell-type-specific origins of the heterogeneity in reinforcement responses.”

“Although a substantial proportion of all VIP cell’ boutons directly target excitatory neurons (Kullander and Topolnik, 2021; Zhou et al., 2017), and the CCK^+^ VIP cells (a smaller subpopulation than the interneuron-specific cells) are known to primarily innervate them (Guet-McCreight et al., 2020), at the functional level, reinforcer-induced activation of VIP cells, as a population is likely to produce a net disinhibitory effect (Lee et al., 2013; π et al., 2013) and thereby gain modulation (Pi et al., 2013) through changing the balance of inhibition across the somato-dendritic axis (Pfeffer et al., 2013).”

“… As VIP neuron subtypes can have diversified functional roles in the microcircuit beside disinhibition (Kullander and Topolnik, 2021), future research should also focus on these sophisticated circuit operations.”

2. Was there a small fraction of VIP+ cells activated by all tasks (Hit, Miss, CR and FA; Figure 1C)?

Figure 2C shows that 91 neurons were activated during all 4 trial types. This was, however, not described in the previous version of the manuscript. We now say in the Results section:

“…12% of the cells were activated in all trial types.”

Not sure, why the AVG response/area is shown. As different cell types were sampled, the best would be to arrange them by depth/layer using the full advantage of the fast volumetric imaging (as shown in Figure S2).

We thank Reviewer #1 for the constructive idea. We wanted to emphasize in Figure 1C the high response ratio of the VIP interneuronal population in Hit and FA, this was the reason why we ordered the responses according to their maximum amplitude. Our analysis showed that responsiveness does not depend on subtype specificity, therefore we think that it is still rational to depict the transients of the entire VIP population together. But we agree that the suggested depth-dependent arrangement could be informative so we have rearranged the transients by depth and added the new panels to Figure 1—figure supplement 1B.

Also, cells could be arranged by their soma size given that CR+/VIP+ have a smaller soma when compared to VIP basket cells (Luo et al. 2020).

Similarly to the previous point, this is an excellent idea. We have added the figure panel with the aforementioned arrangement to Figure 1—figure supplement 1B.

3. TCA allowed authors to understand the variability in the activity dynamics due to inter-trial variability. The interpretation of the obtained results should be added to the main text (Figure 3H) for better understanding.

We thank the reviewer for pointing this. We have added the interpretation of the TCA into the Results section:

“TCA allowed us to quantify, on the one hand, the brain-wide recruitment of VIP interneurons by reward- and punishment-associated responses and, on the other hand, brain region-dependent heterogeneity in dynamics (e.g., slower dynamics in the visual cortex) in an unbiased manner”

4. VIP interneurons from all regions under investigation but the auditory cortex showed arousal and locomotion modulation of reinforcement-mediated recruitment of VIP interneurons. The possible reasons need to be discussed in the main text. Similarly, the absence of arousal (pupil diameter change based) modulation of reinforcement response for Hit & FA trials (unlike Miss & CR trials) requires explanation.

Thank you for raising this point. First, we should note that the effects of running do not seem to be mediated by the VIP network in this cortical region (Yavorska et al., 2021), which agrees with our findings. We are uncertain about why VIP cells in the auditory cortex show less arousal modulation than those in other brain areas, but there is one mechanism that could possibly explain our findings. Similarly to lateral inhibition in V1 by SOM neurons, there is a tone-related network suppression in the auditory cortex (Kato et al., 2017). Interestingly, the intensity of the suppression depends on the arousal: it is weaker at higher arousal levels. As we mentioned in the supplementary information, when the baseline pupil diameter was small, the increase in pupil diameter tended to be higher, and vice versa (Figure 3—figure supplements 1A and B). It means that in trials with a high pupil change the baseline arousal was generally lower. Indeed, in Hit and FA the average peak amplitude of the low baseline pupil group was 1.9-fold larger than in the high baseline pupil group (Author response image 1).

**Author response image 1. sa2fig1:** Comparison of peak amplitudes of pupil trace averages split according to their baseline amplitudes in Hit & FA.

Therefore, we can say that in the high pupil change trials, the network suppression could have a large effect, meaning that the SOM cells could successfully inhibit VIP activity after the tone onset, therefore the arousal change related activity could be reduced.

The reason why the arousal change modulation is still present in Miss and CR trials (Author response image 2) is that the ratio of the average peak amplitude of low and high baseline pupil is smaller, meaning that the high pupil change was less straightforwardly coupled with low baseline pupil, so the effect of the network suppression was reduced.

**Author response image 2. sa2fig2:** Comparison of peak amplitudes of pupil trace averages split according to their baseline amplitudes in Miss & CR.

Importantly, the difference between low and high arousal change was one magnitude larger in the case of Miss & CR (considering the peaks: 14-fold) than for Hit & FA (2-fold).

**Author response image 3. sa2fig3:** Comparison of peak amplitudes of pupil trace averages split according to the pupil change in Miss & CR and Hit & FA (see Figure 3—figure supplement 1A).

Summarizing the aforementioned factors, the tone-related network suppression could be accountable for reduced arousal change modulation in the case of Hit & FA, but the difference between low and high pupil change groups was so high in Miss & CR that the network suppression – that was already weaker in Miss & CR – could not overwhelm this modulation in these trial types. We have strengthened our discussion regarding this point:

“The lack of modulation by arousal change in the auditory cortex can be explained by the arousal-level-dependent, tone-induced network suppression (Lin et al., 2019), a mechanism that in a similar way to lateral inhibition in V1 is controlled by SOM interneurons (Kato et al., 2017). High pupil change trials are usually associated with low baseline arousal and thereby with a high level of network suppression, reducing the activity of the VIP cells that is connected to pupil change and, as a consequence, diminishing pupil change – VIP cell activity correlation. As the connection between high pupil change and low baseline pupil was weaker in Miss & CR trials – and also the difference of high vs. low pupil change was higher than in Hit & FA (Hit & FA: ~2-fold, Miss & CR: ~14-fold, considering the peaks) – the effect of the network suppression did not prevail in these trial types.”

5. Fiber photometry recordings from the auditory cortex (Figure 2A) reveal higher calcium transients for Hit and FA compared to CR and Miss trials, however, the data from Figure S4A shows higher ΔF/F (%) for Miss & CR compared to Hit & FA. Whether the results in Figure 2A and Figure S4A were obtained from different sets of mice should be mentioned.

The data were obtained from the same mice. Unfortunately, in Figure 2A the baseline fluorescence values were calculated in the wrong time interval in this particular case, which shifted the calcium responses: we have now corrected this.

Indeed, although the peak of average FA signal was higher than Miss & CR signals, the average ΔF/F (%) for Miss & CR was larger during the 0-2 s interval than for Hit & FA. This time window was chosen for analysis in order to capture the slow time course of the calcium signals.

6. Authors conclude in the Discussion section that – only the late response phase showed a correlation with the pupil size, whereas the initial, transient phase followed more closely the reinforcer delivery, however in the Figure 3 D-E, ΔF/F (%) for Hit & FA from high vs low pupil size change trials in the late response phase (~2-3 sec from the outcome) show roughly similar responses (especially in motor cortex Figure 3D right). The clarification of how the initial and late response phase was defined is necessary for better understanding. A graph comparing the responses (ΔF/F) of VIP interneurons for high vs low pupil size change trials in initial vs late phase would clarify the extent of the influence of arousal on reinforcement-mediated recruitment of VIP interneurons. Moreover, comparison of ΔP/P in Figure 3C and Figure S4A reveals that reinforcement stimuli trials (FA & Hit) lead to higher pupil size change compared to Miss & CR trials. Considering the modulation of VIP recruitment due to arousal (pupil size change), the reinforcement-mediated recruitment of VIP interneurons might be strongly influenced by arousal. More clarification is needed to conclude about reinforcement-mediated VIP recruitment.

If we complete Figure 3C with a bar graph showing average ∆P/P (%) in high and low arousal change in Hit & FA together, and we compare these bars with the calcium responses Figure 3D, we can see that the differences between the high and low arousal groups are not proportional (~5-fold difference in ∆P/P, ~1.5-fold in ∆F/F, red double-headed arrows in Figure 3), suggesting that a large part of the reinforcement-associated response below the dashed blue line is independent of arousal.

We performed multiple analyses to strengthen this part of the manuscript. As suggested, we have added a panel comparing the difference between high and low arousal-related calcium signal in the initial (0-1 s after reinforcement) and late (2-3 s after reinforcement) response phase: these show an approximately 2-fold larger difference in the late response phase (initial: 10.7 ± 0.01% late: 19.9 ± 0.01%). We have rephrased our conclusion in the Discussion section accordingly.

We now say in the results:

“This modulation was stronger in the late response phase (difference between large vs. small pupil at initial [0-1s] and late [2-3s] phase ΔF/F: 10.7 ± 0.01% vs. 19.9 ± 0.01%).”

We now say in the discussion:

“One caveat of comparing reinforcement-evoked responses to arousal modulation is that the delivery of water reward and air puff punishment usually also drives additional changes in arousal, leading to pupil dilation and/or locomotion. However, the relative change in pupil diameter between the small and large dilation groups was larger than the relative change in the corresponding average reinforcement responses indicating that reinforcement-associated change has a large, arousal independent component (Figures 3C and D). Reinforcement also drove VIP activity when additional pupil dilation and/or locomotion was not detectable. Nevertheless, we found that VIP interneuron recruitment by reinforcers was correlated with pupil dilation, similar to previously-documented arousal modulation (Fu et al., 2014; Garcia-Junco-Clemente et al., 2017; Jackson et al., 2016; Reimer et al., 2014). The late response phase showed a stronger correlation with the pupil size, whereas the initial, transient phase followed the reinforcer delivery more closely (Figure 3—figure supplement 1E).”

Finally, we estimated the distinct contributions of arousal, locomotion, and reinforcement in a systematic way with the help of a generalized linear model (Figure 3—figure supplement 2D, see response to Reviewer #3 for details), which revealed the high relative contribution of the reinforcement to the VIP activity.

7. In the Discussion section (line 440-441), the authors claim that the VIP activity modulation due to locomotion is weak, but the comparison of activity change due to arousal (Figure 3 C-E) and locomotion (Figure 3 F-H) shows that the ΔF/F (%) values in all brain regions for arousal and locomotion modulation are comparable. The claim of weak locomotion modulation of VIP activity is not supported by the results. Also, speed modulation was assessed at a very low animal speed (avg 1.2 cm/s, which was used as a high speed). Have authors tried to look at higher speed locomotion, which has been used in the previous studies?

Our major goal was not to assess VIP activity with locomotion per se, but rather to show that reinforcement signals are impacted, but couldn’t be explained, by locomotion – see for example Figure 3F and G low speed-change FA traces, where the average speed change is 0 or slightly negative (meaning the animals slow down), yet the reinforcement-related activity remained intact. We inserted Author response image 4 to illustrate the different locomotion profiles of the mice and how these relate to average calcium signals. If the animal was still, the correlation between locomotion and VIP responses was ~0, as shown in the example (see the red mouse in Author response image 4). In contrast, the correlation between average pupil size and calcium level was higher, therefore the distribution is shifted towards more positive values, and it was also 2-fold wider (compare Figure 3B to Author response image 4). We should note that the pupil vs. calcium and movement vs. calcium correlations (Author response image 4 and Figure 3B) were calculated over a longer time window, whereas bar plots of Figure 3D-H show the average calcium levels in the 0-2 s time window after the reinforcement onset. Indeed, in this shorter temporal interval, which focuses on the time of reinforcement, the speed and pupil modulations are comparable (only slightly higher in case of the pupil, compare Figure 3D with G and E with H). In summary, this means that on average VIP activity correlated better with arousal than running, but if we restrict our analysis to the reinforcement time interval, the correlations are similar.

**Author response image 4. sa2fig4:** Left, correlation of locomotion and VIP activity (median=0.11). Right, three example mice with low (red) and high (purple and green) running speeds. Right top, average running speed during Hit & FA. Right bottom, corresponding average VIP activity.

Results produced by the generalized linear model also supported our original claims (Figure 3—figure supplement 2D, modulation by locomotion is weaker). Our experience, both in Budapest and Cold Spring Harbor, was that the VIP-cre animals are really calm and didn’t run much, especially compared to PV-cre or Thy1-cre mouse lines. Given this, and as our primary focus was not to measure locomotion correlation, we haven’t tested higher speeds.

In the Results section we now say:

“However, the correlation was weaker at the single-cell level (median correlation coefficient 0.11).”

In the Discussion section we now say:

“…our behavioral paradigm did not encourage mice to run, and their small movements produced only weak modulation in VIP activity (pupil: 0.31, locomotion: 0.11) when correlation coefficients were calculated over a long temporal interval. But modulation was similar when restricted to the reinforcement time interval (0-2 s, Figure 3).”

8. The authors need to discuss the potential mechanisms of co-activation of VIP+ cells (lines 169-173).

We have added a possible explanation to the discussion:

“Cholinergic activity in the neocortex was necessary for learning and modulated VIP cells during locomotor activity (Ren et al., 2022), also potentially being responsible for the co-activation in the auditory discrimination task.”

Reviewer #2 (Recommendations for the authors):– At least for one area (e.g. MPta) the interindividual variance of VIP+ interneuron activity to the different trials and responses has to be provided. This would importantly add to the understanding of the sources of heterogeneity in the recruitment of VIP+ interneurons.

The inter-individual variance of VIP+ interneuron activity of FA and Hit responses is shown in Figure 2—figure supplement 1A but with no detailed analysis.

Therefore, following the reviewer’s recommendation, we performed further analysis and found that, in line with our PCA results, reinforcement response variability (measured as the coefficient of variation, CV) was not primarily attributable to inter-individual variability as it was ~2.6-fold lower than cell-to-cell variability (in Hit and FA). However, in Miss and CR the CV values were surprisingly similar. We explain these findings in the Results section and have added a new panel to Figure 2—figure supplement 2:

“Beyond cell-to-cell and inter-trial variability, inter-individual (animal-to-animal) variability can also contribute to the heterogeneity experienced in the responses. Therefore, we calculated the coefficient of variation (CV=SD/mean) of the peak of responses among the neurons of individual recordings (cell-to-cell) and compared the result to the CV calculated using the average peak amplitudes of the sessions (inter-individual). We conducted this analysis in parietal cortical measurements (n=4 mice, n=236 cells) separately in the four trial types (Figure 2—figure supplement 2E). The cell-to-cell CV was comparable among the trial types, Hit and FA being slightly less variable (Miss: 0.80, CR: 0.70, Hit: 0.57, FA: 0.58). The inter-individual variability in Miss and CR was similar in magnitude to the cell-to-cell variability, but in Hit and FA it was, surprisingly, ~2.6-fold lower (Miss: 0.94, CR: 0.69, Hit: 0.25, FA: 0.20). This means that while the response to tone (without reinforcement) was as variable at the level of individual mice as at the level of individual cells, the reinforcement-related response was less variable across animals, and differences rather came from the cellular level. This latter statement is in good agreement with the PCA results, where the separation did not happen according to the imaging section (Figure 2—figure supplement 2A).”

– The stability of the VIP+ interneuron response to Hit and FA across trials should be shown.

We performed an analysis to check the stability of the VIP interneuron response. We define stability as a portion of active neurons in a trial. As shown in the figure below, although there was some fluctuation, the stability was maintained across trials. We added this panel to Figure 2—figure supplement 1.

Reviewer #3 (Recommendations for the authors):A discrepancy exists with published data that do not show large responsiveness of VIP neurons to reward in visual and barrel cortices. The authors hypothesize that VIP neurons might only respond to reward and punishment during learning. This is the reason they use mice in the phase of task acquisition, that still have "enough" FA trials, while the published studies measure VIP activity in overtrained mice. To solidify this suspicion, the authors could show that, once their mice have fully mastered the task, the reinforcer responses in VIP cells are in fact reduced or absent. This could harmonize their results with these published studies and significantly strengthen the paper.

This is a great point and actually true. We are working on a second manuscript addressing that in detail – see response to editor above.

However, in our opinion, data in Figure 2—figure supplement 2D already support this view: we show there and in the connected linear regression analysis the relative size of the average cue response compared to the average reinforcement response as a reference (as described in the methods), as a function of the hit rate, which is a behavioral performance parameter. ∆F/F_tone_ is larger when the hit rate is higher, and the hit rate in our best-performing animals (70-79%) was just slightly lower than that found by Khan et al., 2018, Sachidhanandam et al., 2016 (~85% accuracy, hit rate>80%). We have added a sentence to the corresponding part of the discussion for clarification:

“Those studies, however, used over trained animals for which little to no punishment was delivered and reward delivery was fully predictable. One study limited to the amygdala indeed showed that reinforcement recruits VIP neurons in a time-limited manner (Krabbe et al. 2019). In line with that, in our measurements from well-performing animals, the cue component dominated the signal (Figure 2—figure supplement 2D).”

Figure 1: Why are the averages in Figure 1 C and D not corresponding? From the legend it appears all curves should be the averages of the above shown 120 VIP cells. Yet their relative amplitude, as well as time course is definitely different. The hit-curve in 1C seems to start to rise before reward delivery, yet this does not seem to be the case in 1D. In 1D it is unclear what the CR and miss trials are aligned to. Is there really error bars on the bottom plots in 1C as mentioned in the legend?

Unfortunately, the legend of Figure 1 was not specific enough. Figure 1C top shows individual trials for Miss, CR, Hit, and FA from the 120 VIP neurons recorded simultaneously. The bottom 4 traces are the corresponding averaged activity of the 120 cells during the 4 individual trials. In contrast, Figure 1D indicates averages of 120 cells and 128 trials (session averages). We have now clarified this information in the legend. This also explains why the mean ± SEM values are not visible: mean, mean ± SEM transients overlap at this high n number (n = 120 × 128 = 15,360). We use thinner lines now and a darker shade of gray, therefore SEM values are more easily distinguished. We have also magnified the traces and aligned the baseline. In panel D Miss and CR traces were aligned to the expected onset of the reinforcement delivery, calculated as the sum of the time between triggering and receiving the reinforcement (fixed 0.5 s) and the time elapsed between stimulus onset and the lick that triggered reinforcement (reaction time). We used the same method to align Miss & CR trials in Figure 2A.

In Figure 1D legend we now say:

“Left, average transients of a measurement session (128 trials) for Hit (green), FA (red), Miss (dark blue), and CR (light blue) responses recorded from the 120 VIP interneurons. Gray triangle marks the reinforcement onset in case of Hit and FA. Averages of Miss and CR trials were aligned according to the expected reinforcement delivery calculated based on the average reaction time. Right, average synchronicity (mean ± SEM) and trial-to-trial repeatability (reliability) of individual neuronal responses.”

Was MPta chosen for this figure simply because it yielded the biggest n? In Figure 2 it appears to almost be an outlier of all the imaged areas, as it is one of very few areas where VIP neurons appear to entirely lack sensory responses.

The cortical location of the imaging site did not significantly affect the size of the cue response. For example, in our other Mpta recording, VIP cells were recruited mainly by the cue onset (Figure 2A). Consistently, we did not find any area-specific modulation of the sensory response within the dorsal cortex (p=0.41, see Behavioral performance influences task-related VIP interneuron responses), the hit rate is a better determining factor as it can cause ~5-fold change in the amplitude of the tone-associated component of the response. For better visibility we have made the arrowheads in Figure 2A smaller and used thinner traces to show the cue and reinforcement components more clearly.

However, it is true that we selected this particular Mpta measurement because of the high cell number and the high SNR, as it can demonstrate the main advantages of our random-access scanning system, recording sparsely-labeled interneuron populations from a large volume with good temporal and spatial resolution. As in Table S1 we made a theoretical comparison between imaging systems with different operation principles: it looked reasonable to represent this technical aspect on the first figure too.

Figure 2: In panel A it is exceedingly hard to distinguish the thin lines from the same color thick lines and thus see potential sensory responses in any of the areas, perhaps one could change the hue slightly or use the different shade of blue as in panel H? In general, a different color scheme could be considered for accessibility reasons. And a legend should be added.For the speaker sign at the bottom of B it is unclear where the start of the auditory cue is, and if it is the same for all the lines of a plot.

We thank the reviewer for the constructive comment. We have now used a different color code for Miss and CR and have modified the legend accordingly. Regarding the auditory cue, it is important to note that FA- and Hit-associated transients were aligned to reinforcement onset before averaging. Therefore, the cue onset time could vary from trial-to-trial covering a narrow interval (for example in fiber photometry measurements, reaction times in Hit and FA trials were 0.35 ± 0.147 s and 0.84 ± 0.108 s). To clarify the interpretation, we have added this information to the figure legends. In addition, we now use thinner lines for Hit and FA for better visibility.

The time courses here, as well as the ones in F make it seem as though in this data set there was a fixed interval between cue presentation and reinforcer delivery. Or is it similar to Pi et al., where the animals triggered the reinforcer by licking. This is not mentioned anywhere.

In this aspect our task was identical to the one used in Pi et al., 2013: the animal had to lick to receive reward, thus the interval between tone and reinforcement onset varied from trial to trial (for average traces aligned to the tone onset see Figure 2—figure supplement 2C about time warping).

Accordingly, in Figure 2A, speakers indicate the average and in Figure 2B the approximate time of tone onset in the Hit and FA trials. However, there was a difference between the 2-photon and the photometry setups: for photometry, the tone stopped immediately after the lick that triggered reinforcement. We have now added this information to the methods section and the figure legend.

In the methods section we now say:

“The absence of licking in go trials was not rewarded (Miss trials). If the animal correctly withheld licking to no-go tones (correct rejection, CR), the air puff was omitted. The time interval between tone and reinforcement onset was a function of the reaction time of the animal. In the figures, responses were aligned to reinforcement onset before averaging in FA and Hit which resulted in a jitter in cue onset time. Reinforcement came 0.5 s after it was triggered, except for during the fiber photometry recordings where reinforcers were delivered upon licking (see below).”

“The photometry experiment used the updated version of Bpod that allows the tones to stop immediately when a mouse licks during the tone. Therefore, the duration of tone in the Hit and FA trials varied. Note that when the behavioral task was set up for the two-photon experiment, this function was not available in Bpod and the tone duration was fixed to 0.5 s for all trial types.”

In Figure 2F, we applied PCA on the period from 0-4 s after reward delivery (see methods) in Hit trials to quantify sources of variance of the reward-related response. As the tone response occurred before this period, this component appears to be more homogenized across the clusters, especially because these clusters reflected cell-to-cell rather than animal-to-animal variance (compare Figure 2—figure supplement 2A and Figure 2F tSNE plots and see also the new analysis of inter-individual variance and the related Figure 2—figure supplement 2E). In addition, Figure 2F contains only data from cells that were significantly responsive in Hit trials, therefore the heterogeneity is already reduced compared to Figure 2A, as non-responsive cells are not depicted.

One session in M1 (hits), as well as one session in M2 (FAs), VIP neurons seem to respond before the onset of the auditory stimulus (judging from the other areas time courses), why is this?

As we explained in the previous point, FA and Hit responses were aligned to the reinforcement onset time. The small jitter (~100 ms SEM) in the reaction time blurred the rise of the cue-associated component during averaging which resulted in a slowly rising, ramp-like component. Therefore the speaker sign in Figure 2B indicates the approximate, and not the exact, onset of the auditory stimulus. In the case of the M1 and M2 measurements referred to, both reaction time and jitter of the reaction time was ordinary, but a small positive trend was observed in the baseline period that resulted in a relatively large color change with the previous improper color settings, which we have now corrected.

All abbreviations should be explained in the legend (esp. panels F,G,H)

Corrected.

Figure 3 C,D,E: why were these areas chosen, or which areas are included in each of these plots? Do differences between hits and FAs exist? There appear to be potentially interesting areal differences.

We selected mPFC, SS, and Mtr, because we wanted to choose regions which are relevant for our behavioral experiments. Measuring ACx is evident, as the cue was a tone, mPFC is involved in task-dependent choices and reward representation, the motor cortex is required for the motor reaction; we also added the somatosensory cortex, as it is also inevitably activated by tactile sensory inputs during the task.

Figure 3C represents the average pupil change during the measurements from the motor and somatosensory regions (Figure 3D). The corresponding pupil traces of Figure 3E (mPFC) can be found in Figure 3—figure supplement 1A:

A. Average pupil dilation traces during high (red and orange) and low (black and gray) pupil changes for FA and Hit trials for SS and Mtr recordings in panel D.

B. Population averages for Hit and FA during high and low pupil change in the SS and Mtr regions. Bars indicate average peak amplitudes (mean ± SEM, Hit and FA combined). Even in the late period, when the outcome responses were dissipated, larger changes in pupil diameter at the time of reinforcement were associated with higher VIP responses.

C. Same as D but for fiber photometry in the mPFC. Corresponding pupil dilation traces can be found in Figure 3—figure supplement 1A.

Indeed, we observed some trial type-specific differences (Figure 3—figure supplement 1C), but our general observation was the presence of modulation across the cortex, except for the ACx.

In the figures of the time courses of the ACx repsonses (2A and S1D) it seems like VIP cells only have sensory responses in miss and CR trials. Why is this bump absent from hit and FA trials?

In our task design in the photometry experiment, if a mouse licked during go or no-go tones, the tones stopped immediately and a reward or punishment was delivered. The reaction time of mice in the Hit and FA trials were 0.35 ± 0.147 s and 0.84 ± 0.108 s, respectively. Considering the slow response time of GCaMP (compared to electrical signals) and photometry collecting population signals, it is a short time window for a bump to be developed. After a discussion and feedback from our colleagues, we decided to align the calcium traces of Miss and CR to the mean reaction time and added an explanation to the legend of Figure S1D, which is now Figure 1—figure supplement 1E. Traces in Figure 2A were also adjusted accordingly. We hope this alleviates any potential confusion for readers.

line 227: How are these 606 reward delivery sensitive cell related to the 622 in Figure 2?

We focused on reaching low reconstruction error, so removed 16 cells with lower SNR that would have disproportionately increased our error. We added this information to the methods section:

“… 16 cells were excluded from this analysis as they disproportionately increased the reconstruction error.”

For the arousal and running modulation: why is the data split here and no correlation calculated as for the relationships to behavioral performance and sensory tuning?

We reasoned that splitting the data by the median value of pupil change / baseline pupil diameter and speed change could better highlight the modulatory effects of the arousal and locomotion compared to showing correlations only. However, we calculated correlation between cell activity and pupil size and the median value was 0.31 (see line ~346 and Figure 3B). We also calculated the correlation with the locomotion speed, and we added the median value (0.11) to the corresponding section (see line ~397)

The running paragraph: it needs to be clarified whether they look at running speed per se, as much of the paragraph is written (fast and slow running conditions), or at speed changes, as the beginning of the paragraph states and the values (partly negative) would make it seem. If it is indeed speed change, they should rename the conditions.

We thank the reviewer for the helpful comment. Indeed, we show speed changes (∆speed) in Figure 3 and Figure 3—figure supplement 1. We replaced fast and slow running phrases with high vs. low speed-change. We describe how speed change was calculated in the methods:

“Velocity traces were first Gauss filtered. They show the absolute speed of the movement, regardless of the direction. In Hit and FA trials, we defined the change in the running speed as the speed difference between the reinforcer delivery time period (0-2 s interval after reinforcement onset) and baseline time period (−2-0 s interval before tone onset). In Miss and CR trials, the speed difference was calculated between the tone delivery time period (0-2 s interval after tone onset) and baseline time period (−2-0 s interval before tone onset). Trials were separated into low and high speed-change groups according to the median speed-change value.”

Figure S2F seems very important. Was also uncued negative reinforcement presented? How does the uncued data compare to the cued?

Uncued air puff wasn’t presented during the auditory discrimination task, however we collected data from a Pavlovian task, in which both cued and uncued reward were presented. We compared the responses in Author response image 4.

Is S5A the exact same data as in 4D, just plotted in a different way? A statistical test should be performed to judge the significance of this correlation.

Yes, the two plots represent the same data. The correlation of the reinforcement- and visual stimulation-related responses turned out to be significant. We thank the reviewer for pointing this out. This correlation could come from the differences in general excitability.

The methods need to be improved. As is there is not enough detail and analyses cannot be understood properly. For example: What does smoothing according to the reaction time periods mean? Reaction time periods are not even mentioned in the description of the task. Please explain time-wrapping. Why is 1.5s 30 data points when the sampling rate is 27.7 Hz and not 20Hz? Why are only 10 trials selected? Why the first? Why is K then 40?

We have now improved the TCA-related methods section according to the reviewer’s comment. As for the specific points raised by the reviewer:

1. The single-trial traces were smooth using the MATLAB smooth function

2. The sampling rate of our AOD-2p recordings varied depending on the number of cells being recorded. For the TCA, all trials were aligned to the start of the cue delivery. However, because of the varying sampling rate, the subsequent delivery of the outcome (Hit and FA) does not align across sessions. The jitter in reaction time also resulted in variance of the recorded frames across trials (see more about the reaction time in the “Auditory discrimination task” section of methods). Applying TCA or any type of clustering without fixing these sampling rate and reaction time related jitters could lead to a classification according to the sampling rate and not to the trial types or other task-related event. Thus, we decided to mitigate this risk by ‘time-warping’ the period corresponding to the cue delivery as described in Williams et al., 2018 (i.e., down- or up-sampling to 20 Hz and fixing the time between tone reinforcement delivery to 1.5 s).

3. The number of trials of each type varies across sessions. To build an N×K×T matrix where N is the number of neurons coming from different sessions, K the trial numbers (10 per trial type = 40 total) and T the time, we had to select the same number of trials for each neuron coming from different sessions. We used 10 trials per trial type per session/neuron as it allowed us to include all sessions (i.e., all neurons) while maintaining an equal representation of all trial types. Our rationale for using the first 10 trials of each trial type was that it would limit the variability in the animal states. However, we did try using 10 trials per trial type at random and observed the same effect.

We now say in the discussion:

“We applied Tensor Component Analysis (TCA) (Kolda and Bader, 2009; Williams et al., 2018) on somatic ca^2+^ responses recorded during the discrimination task. After smoothing, single-trial neural activities corresponding to the reaction time periods for Hit and FA trials were time-wrapped to a fixed 1.5 s / 30 data points in length. This step was necessary because: 1, the reaction time varied from trial to trial; and 2, the variable number of neurons recorded in the different sessions led to a variable sampling rate for imaging. These factors led to a trial- and session-dependent number of images between the cue and the reward delivery. All recordings were rendered non-negative by subtracting the minimal fluorescence ΔF/F value for each cell. Data were finally normalized by dividing by the average maximum fluorescence ΔF/F value of Hit-only trials. Only the first 10 trials of each type were then selected for each cell and each session and assembled in an N×T×K matrix where N=771 neurons, T=time (s), K=40 trials. TCA reconstruction error was computed with different latent numbers [1,2,3,4,5,10,15,20] with 10 different initial conditions for each latent number. Using 3 latents led to a reconstruction error of 21%.”

Supplementary figure S3C is never mentioned.

Corrected. We mention it now in the section about the TCA.

“…After preprocessing (Figure 2—figure supplement 2C) we used non-negative tensor decomposition (Kolda and Bader, 2009; Williams et al., 2018) and focused our analysis on rank 3 TCA…”

We thank Reviewer #3 for all the constructive comments.